# RECON: ROBUST SYMMETRY DISCOVERY VIA EXPLICIT CANONICAL ORIENTATION NORMALIZATION

**Alonso Urbano**[1]* **David W. Romero**[2] **Max Zimmer**[1] **Sebastian Pokutta**[1,3]

[1] Department for AI in Society, Science, and Technology, Zuse Institute Berlin (ZIB), Germany
[2] Cartesia AI, San Francisco, CA, USA
[3] Institute of Mathematics, Technische Universität Berlin, Germany

{urbano,zimmer,pokutta}@zib.de dwromerog@gmail.com

## ABSTRACT

Real world data often exhibits unknown, instance-specific symmetries that rarely exactly match a transformation group $\mathcal{G}$ fixed a priori. Class-pose decompositions aim to create disentangled representations by factoring inputs into invariant features and a pose $g \in \mathcal{G}$ defined relative to a training-dependent, *arbitrary* canonical representation. We introduce RECON, a class-pose agnostic *canonical orientation normalization* that corrects arbitrary canonicals via a simple right translation, yielding *natural*, data-aligned canonicalizations. This enables (i) unsupervised discovery of instance-specific pose distributions, (ii) detection of out-of-distribution poses and (iii) a plug-and-play *test-time canonicalization layer*. This layer can be attached on top of any pre-trained model to infuse group invariance, improving its performance without retraining. We validate on images and molecular ensembles, demonstrating accurate symmetry discovery, and matching or outperforming other canonicalizations in downstream classification.

| Data | Class-pose Canon. | Class-pose Distrib. | RECON Canon. | RECON Distrib. |
|---|---|---|---|---|
| $\{ \mathcal{1}, .. 7, .. \mathcal{7} \}$ | $L$ | $\nu_{[7]}$ | 7 | $\mu_{[7]}$ |
| $\{ \mathcal{5}, .. 5, .. \mathcal{S} \}$ | $\mathcal{5}$ | $\nu_{[5]}$ | 5 | $\mu_{[5]}$ |

(a) Inputs with identical $\pm 30°$ symmetries.

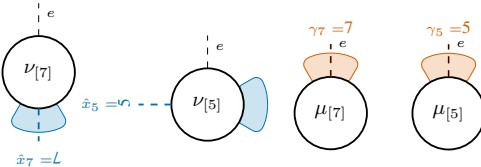

(b) Relative distributions $\nu_{[x]}$ (□) vs normalized distributions $\mu_{[x]}$ (□) via RECON.

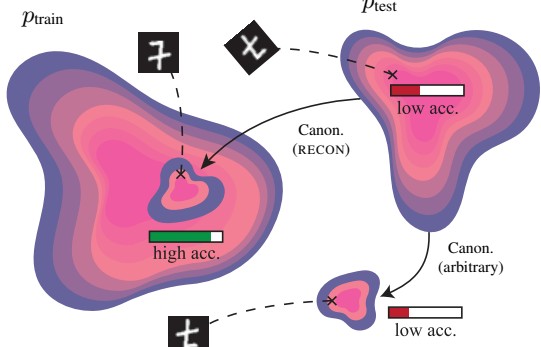

(c) Distribution shift ($p_{\text{train}}$ vs $p_{\text{test}}$) induced by unseen symmetries at test time, corrected by RECON.

Figure 1: **(a)** Class-pose methods assign arbitrary (often out-of-distribution) canonicals per class. **(b)** This leads to distinct relative-pose distributions $\nu_{[x]}$, obscuring the shared $\pm 30°$ symmetries. RECON corrects these offsets, mapping inputs under the same symmetries to the *same* distribution $\mu_{[x]}$ and extracting their *natural pose* $\gamma$. **(c)** Our data-aligned canonicalization removes symmetry-induced distribution shifts, improving downstream performance of pre-trained backbones without architectural restrictions or retraining.

## 1 INTRODUCTION

Symmetry transformations like rotations arise naturally in many domains (Cohen & Welling, 2016; Higgins et al., 2018; Bronstein et al., 2021), with objects appearing in poses related by group transformations $g \in \mathcal{G}$ (e.g., molecules in different orientations) (Weiler et al., 2018; Brandstetter

---

* Corresponding author: urbano@zib.de

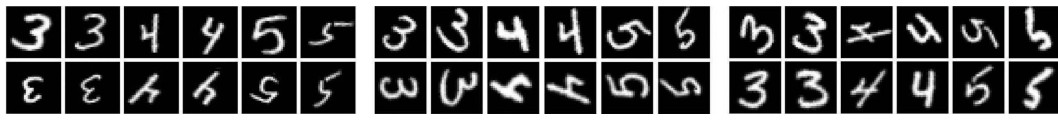

|  (a) SGM  |  (b) IE-AE  |  (c) RECON (Ours)  |

Figure 2: Comparison of canonical representations. Top row: Input MNIST digits under varying rotations. Bottom row: canonicals generated by (a) SGM(Allingham et al., 2024) and (b) IE-AE (Winter et al., 2022), which yield arbitrary orientations, while RECON consistently produces data-aligned canonical poses (upright digits) (c).

et al., 2022). While $\mathcal{G}$-equivariant neural networks exploit such structure (Cohen et al., 2018; 2019; Romero et al., 2020; Romero & Cordonnier, 2020; Wang et al., 2020), they can be overly constraining when there is a mismatch between the pre-fixed group $\mathcal{G}$ and the symmetries in the data (Weiler & Cesa, 2019; Romero & Lohit, 2022). In effect, real-world symmetries are often (i) unknown a priori, (ii) partial (covering only part of the group), or (iii) instance-dependent. This motivates methods that *discover* symmetries from data (Benton et al., 2020; Forestano et al., 2023; van der Linden et al., 2024; Allingham et al., 2024).

However, existing approaches often require supervision or learn only dataset-level patterns (Sec. 4). Our goal is to discover, in an unsupervised manner, the group transformations inherent to each instance; in particular, we aim to learn probability distributions on $\mathcal{G}$ that describe the poses in which each instance appears in the data.[1] We argue that a promising foundations lies on class-pose decomposition methods (Winter et al., 2022; Yokota & Hontani, 2022; Marchetti et al., 2023; Allingham et al., 2024), which disentangle inputs into invariant *class* features and a *pose* $g \in \mathcal{G}$ relative to a learned canonical representation. The canonical's orientation is generally *arbitrary* (Winter et al., 2022; Allingham et al., 2024), which results in an out-of-distribution (OOD) canonicalization and in arbitrarily shifted relative-pose distributions (Fig. 1).

We address arbitrary canonicals and propose a framework for *Robust unsupervised discovery of intrinsic symmetry distributions via Explicit Canonical Orientation Normalization* (RECON). We prove (Proposition 3.1) that by estimating the centroid (Fréchet mean) of the observed relative poses, which captures the offset induced by the arbitrary canonical, we can approximate symmetry distributions centered at the input's *natural pose* via a simple right translation. This yields instance-specific symmetry descriptions that are *robust* (independent of the arbitrary canonical), *interpretable* (centered at $e \in \mathcal{G}$, representing the input's natural pose), and *comparable* across classes (Fig. 1b). RECON is validated on 2D image benchmarks and 3D molecular conformations – beyond typical 2D-only settings of prior work (Benton et al., 2020; Romero & Lohit, 2022; van der Linden et al., 2024; Allingham et al., 2024; Kim et al., 2024). Lastly, we provide practical applications in (i) OOD pose detection and (ii) test-time canonicalization, a drop-in method to grant group invariance to frozen pre-trained models, improving downstream performance (Sec. 5.2).

**Contributions**

1. We propose RECON, a method for discovery of instance-specific pose distributions from unlabeled data leveraging class-pose representation learning methods.
2. We achieve this through *canonical orientation normalization* (Proposition 3.1), an architecture-agnostic correction of arbitrary canonicals, yielding data-aligned natural canonicalizations and well-behaved symmetry distributions.
3. We empirically validate on 2D images and large-scale, real-world 3D data. We offer applications in OOD pose detection and test-time canonicalization, offering performance improvements to pre-trained backbones via a simple plug-in canonicalization step.

Our code is publicly available at `https://github.com/ZIB-IOL/recon`.

## 2 PRELIMINARIES

Our approach leverages class-pose decomposition methods, a class of neural networks designed to disentangle input data into an invariant (*class*) component and an equivariant (*pose*) component based on a given transformation group $\mathcal{G}$. In particular, we build upon Invariant-Equivariant Autoencoders

---

[1]Note how this differs from estimating the stabilizer $\mathcal{S}_x$ describing the (self) symmetries of $x$ (cf. Appx D.1).

(IE-AEs) (Winter et al., 2022). We briefly review the core concepts here; formal definitions regarding group theory, representations and $\mathcal{G}$-equivariance are deferred to Appendix A.

IE-AEs aim to learn group invariant and equivariant representations for an input $x \in \mathcal{X}$ of a vector space, e.g., an image or a 3D structure. For a chosen group $\mathcal{G}$, an IE-AE maps $x$ to a $\mathcal{G}$-invariant component $z \in \mathbb{R}^n$ and a $\mathcal{G}$-equivariant group element (pose) $g \in \mathcal{G}$.[2]

**Invariant component $z$ and canonical representation $\hat{x}$**  First, a $\mathcal{G}$-*invariant* encoder $\eta : \mathcal{X} \to \mathcal{Z} \subseteq \mathbb{R}^n$ learns an invariant representation $z = \eta(x)$. This embedding $z$ captures features of the input that are independent of its pose under $\mathcal{G}$, meaning $z$ remains unchanged if $x$ is transformed by any element of the group. A corresponding decoder $\delta : \mathcal{Z} \to \mathcal{X}$ then reconstructs the input $\hat{x} = \delta(z)$ from this invariant representation. Note that since the decoder only sees the invariant embedding $z = \eta(x)$, it will produce the *same reconstruction* $\hat{x} = \delta(\eta(x))$ for all group transformations of the input. This common reconstruction $\hat{x}$ is called the *canonical representation*. Notably, the specific pose or orientation of $\hat{x}$ is *arbitrary*, influenced by initialization or training dynamics (Winter et al., 2022). This behaviour is common to other class-pose decomposition methods (Yokota & Hontani, 2022; Allingham et al., 2024).

**Pose component g**  The canonical $\hat{x}$ has some fixed, arbitrary pose. To recover the original input $x$, we have to determine the specific group transformation $g \in \mathcal{G}$ that maps $\hat{x}$ back to $x$. This is the role of the *group function* $\psi : \mathcal{X} \to \mathcal{G}$, which predicts this transformation $g = \psi(x)$. All the IE-AE components $(\eta, \delta, \psi)$ are then trained jointly by minimizing the reconstruction loss $d(\rho_{\mathcal{X}}(\psi(x))\, \delta(\eta(x)), x)$,[3] where $\rho_{\mathcal{X}}$ is the group action of $\mathcal{G}$ on $\mathcal{X}$.

In summary, IE-AE provides an invariant latent vector $z = \eta(x)$, an arbitrary canonical $\hat{x} = \delta(z)$, and the *relative* group transformation $g = \psi(x)$ that maps the canonical $\hat{x}$ back to $x$. Our method leverages these components, specifically the distribution of relative transformations $g = \psi(x)$ and the invariant latent space $\mathcal{Z}$, to discover the specific distribution of transformations that appear in the data.

# 3 METHOD

## 3.1 PROBLEM STATEMENT

Our goal is to discover the characteristic distribution of symmetry transformations associated with different classes of objects within unlabeled data $\mathcal{X}$. Consider datasets such as GEOM (Axelrod & Gómez-Bombarelli, 2022), where each molecule class is represented as an ensemble of multiple 3D molecular conformations; or MNIST, where same digits share an underlying shape under varying handwriting styles. Generally, variations *within* these classes are a combination of (a) non-group structural distortions (e.g., bond rotations or ring puckering in molecules; style variations in digits) and (b) transformations from a symmetry group $\mathcal{G}$ (like $SO(3)$ rotations of molecular conformations). We specifically aim to model the distribution of the underlying group transformations.

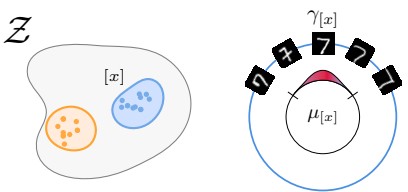

Figure 3: Problem setup. *Left:* A class $[x]$ is defined by inputs clustering together in the invariant space $\mathcal{Z}$. *Right:* We model instances $s \in [x]$ as $\rho_{\mathcal{X}}(g)\gamma_{[x]} + \varepsilon_s$, where $\gamma_{[x]}$ is a reference frame and $g$ is drawn from a distribution over rotation angles $\mu_{[x]}$. The objective is to recover $\mu_{[x]}$ from the unlabeled data.

Therefore, to achieve this goal, we first need to **(i)** define a way to group objects based on their intrinsic shape or similarity – independently of their pose or minor deformations and under the absence of labels, and **(ii)** model their pose variations.

**(i) Modeling pose-invariant similarity**  We model pose invariant similarity through equivalence classes. The $\mathcal{G}$-invariant latent space typical of class-pose decomposition methods is a promising candidate for this purpose. In effect, we rely on the principle that structurally similar objects generally map to nearby points in $\mathcal{Z}$. This assumption is empirically supported by previous work in class-pose decompositions methods showing that $\mathcal{Z}$ maps different classes into well-separated connected clusters (see t-SNE visualizations of $\mathcal{Z}$ in IE-AEs (Winter et al., 2022)). Additionally, it aligns with broader

---

[2]Figure 7 in the Appendix, adapted from Winter et al. (2022), visualizes this architecture.

[3]Any estimator $\psi$ satisfying the proposed reconstruction loss is $\mathcal{G}$-*equivariant* as proved in Winter et al. (2022).

findings indicating that deep features correctly capture perceptual and semantic similarity, even when learned without supervision (Zhang et al., 2018).[4]

Formally, let $\eta : \mathcal{X} \to \mathcal{Z}$ be such a $\mathcal{G}$-invariant encoder, we then define an equivalence relation $\sim_\varepsilon$ in $\mathcal{X}$ based on connected proximity in $\mathcal{Z}$:

$$x \sim_\varepsilon y \iff \exists \{x_i\}_{i=0}^N \subseteq \mathcal{X} \text{ for some } N \in \mathbb{N} \text{ s.t. } x_0 = x, x_N = y, \text{ and } d_\mathcal{Z}(\eta(x_i), \eta(x_{i+1})) < \varepsilon \, \forall i$$

where $d_\mathcal{Z}$ is a norm-induced distance in $\mathcal{Z}$ and $\varepsilon > 0$ is a small threshold. Intuitively, this equivalence relation groups objects whose invariant features form a connected component (Fig. 3, left). We denote the resulting equivalence class containing $x$ as $[x]$.[5] We empirically validate this equivalence class definition in Appendix D.3, confirming effective capture of pose-invariant similarity, and analyze its limitations in detail.

**(ii) Modeling pose variations**   Having established the classes $[x]$, we now focus on modeling the distribution of group transformations responsible for pose variations within each class. We model this as a probabilistic process.

Conceptually, there exists some true underlying probability distribution $\mu_{[x]}$ that governs how instances $s \in [x]$ are generated by transforming some reference pose $\gamma_{[x]} \in \mathcal{X}$ plus residual non-group variations. Formally, let $\mathcal{G}$ be a Lie group and consider $\mu_{[x]}$ a probability distribution over $\mathcal{G}$. We assume that instances $s \in [x]$ can be generated by sampling a transformation $g \sim \mu_{[x]}$ and applying it to a reference pose $\gamma_{[x]}$, plus a deviation term $\varepsilon_s \sim \mathcal{P}_{\varepsilon'}$:

$$s = \rho_\mathcal{X}(g)\gamma_{[x]} + \varepsilon_s, \quad \text{with } g \sim \mu_{[x]}, \, \varepsilon_s \sim \mathcal{P}_{\varepsilon'} . \tag{1}$$

Here, $\mu_{[x]}$ represents the true probability distribution relative to the reference pose $\gamma_{[x]}$. The distribution $\mathcal{P}_{\varepsilon'}$ models variations that are not explained by the group transformation (e.g., style variations in digits, internal conformational changes in ensembles of molecules) and is assumed to have zero mean and small variance (we assume $||\varepsilon_s|| < \varepsilon'$ almost surely for simplicity). Fig. 3 (right) illustrates this model.

Note that under this model, there are several ways of representing the symmetries in the data, depending on the reference pose $\gamma_{[x]}$. Consider a dataset of handwritten '7's exhibiting rotational symmetries uniformly between $\pm 30°$: $\{\text{⌐}, \dots 7, \dots \text{⌐}\}$. Using this model, we can describe the symmetries in this dataset as $\mu_{[7]} = \mathcal{U}([150°, 210°])$ for a reference pose $\gamma_{[7]} = \text{'}L\text{'}$, or as $\mu_{[7]} = \mathcal{U}([-30°, 30°])$ for a reference pose $\gamma_{[7]} = \text{'}7\text{'}$. While both descriptions are *mathematically valid*, the latter, identity-centered representation is far more desirable. It aligns with our intuition of a *naturally occurring pose* and directly reflects the symmetries as deviations from a neutral reference frame ($e$, no transformation), which offers several advantages (as demonstrated in Sec. 5).

**Defining a natural pose**   Ideally, we aim to obtain such identity-centered descriptions of the data symmetries. As we have just exemplified, this boils down to obtaining the symmetry distribution $\mu_{[x]}$ whose $\gamma_{[x]}$ is the "natural" pose in the data – like the upright '7' – centered at the group identity. But without labels defining what a reference or *true* canonical pose is, how can we define a *natural pose*? Specifically, how can we obtain a canonical pose that is geometrically grounded in the symmetries in the data – rather than just an arbitrary pose?

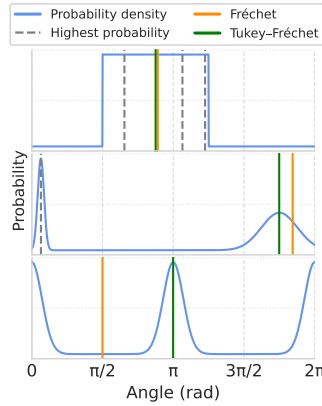

Figure 4: Estimator comparison for different distributions over $SO(2)$.

While the most likely pose (i.e., with maximum probability density) may seem like a good candidate – for instance in the case of Gaussian or other unimodal distributions – it can be ill-defined (e.g., for uniform distributions, see Fig. 4, top) or not representative (e.g., in the case of a distribution with an anomalous sharp spike in its density, Fig. 4, middle). To establish a general definition, we use the *Fréchet mean* (Pennec, 2006) or Riemannian center of mass, a measure of the central tendency of the distribution on the group

---

[4]Structured per-class manifolds in latent spaces have also been shown algebraically (Pelleriti et al., 2025).

[5]While we use this definition throughout our theoretical derivations (see Proofs B), in practice, we approximate the classes $[x]$ by simply computing the $k$-nearest neighbors of $\eta(x)$ in $\mathcal{Z}$ for efficiency (see Algorithm 1).

manifold. Given any distribution $\mu$ over $\mathcal{G}$, its Fréchet mean $\mathcal{F}(\mu)$ is the unique transformation minimizing the expected squared Riemannian distance $d_\mathcal{R}$ to samples from $\mu$:

$$\mathcal{F}(\mu) = \mathrm{argmin}_{y \in \mathcal{G}} \, \mathbb{E}_{g \sim \mu}[d_\mathcal{R}(y, g)^2]. \tag{2}$$

Under mild conditions, the Fréchet mean provides a unique centroid (Afsari, 2011) that aligns with intuition for common cases: it corresponds to the midpoint for symmetric uniform distributions and the peak for strongly unimodal ones (e.g. Gaussian). However, in certain multimodal cases, e.g., a bimodal distribution on $SO(2)$ with opposing identical peaks (Fig. 4, bottom), the Fréchet mean may fall outside the support of $\mu$. This would lead to an out-of-distribution canonicalization, affecting downstream tasks (Sec. 5.2). Therefore, we propose a robust Fréchet mean extension based on robust *M-estimators* (Shevlyakov et al., 2008) that we coin the *Tukey-Fréchet mean* $\mathcal{F}_r(\mu)$:

$$\mathcal{F}_r(\mu) = \arg\min_{y \in \mathcal{G}} \mathbb{E}_{g \sim \mu}[m(d_\mathcal{R}(y, g); c)], \tag{3}$$

where $m(u; c)$ is the Tukey biweight loss (Rousseeuw & Hubert, 2011; Shin & Oh, 2022). This variation extends the Fréchet mean and converges consistently to the same mode in multimodal distributions (cf. Appendix C). For simplicity, we derive our theoretical framework with the classical Fréchet mean, whose properties are well established, but in practice, the robust Tukey-Fréchet can be used as a drop-in replacement.

We thus define our target distribution $\mu_{[x]}$ in equation 1 as the probability distribution over $\mathcal{G}$ that is centered at the group identity $e$ in the Fréchet sense, i.e., satisfying $\mathcal{F}(\mu_{[x]}) = e$. Consequently, the reference pose $\gamma_{[x]}$ (the natural pose) corresponds implicitly to the Fréchet mean (i.e., the upright '7' in the previous example). Our objective is to estimate this well-behaved $\mu_{[x]}$ from unlabeled data. The subsequent section details our method for achieving this by normalizing the arbitrarily offset outputs of IE-AEs.

## 3.2 RECOVERING SYMMETRIES VIA CANONICAL ORIENTATION NORMALIZATION

The core idea is to leverage the disentangled representation $(z, g) \in \mathcal{Z} \times \mathcal{G}$ of the IE-AE to extract the symmetry distribution of an input, and correct for the arbitrary canonical pose to yield a canonical-pose independent distribution. Our main result provides a way to recover the true symmetries $\mu_{[x]}$ from the relative transformations $\psi(x)$. We provide a proof in Appendix B

**Proposition 3.1** (Approximation of $\mu_{[x]}$ via Normalization). *Let $\mathcal{X}$ be a metric space, $\mathcal{G}$ a Lie group and $\eta, \delta, \psi$ an IE-AE where $\psi$ is continuous on a compact domain $\mathcal{X}$. Suppose that $\mathcal{X}$ exhibits symmetries characterized by $\mu_{[x]}$ where $\mathcal{F}(\mu_{[x]}) = e$ as described above. Consider a random sample $\{s_i\}_{i=1}^N$ of $[x]$ and denote their images by $\psi$ as $\psi([x]) = \{\psi(s_i)\}_{i=1}^N$. Let $\hat{\Gamma}_{[x]} \in \mathcal{G}$ be the empirical Fréchet mean of $\psi([x])$. Then, the empirical distribution $\hat{\mu}_{[x]}$ corresponding to the normalized samples $\psi([x])\hat{\Gamma}_{[x]}^{-1}$ approximates the target distribution $\mu_{[x]}$. Specifically, $\hat{\mu}_{[x]}$ converges in Wasserstein distance to $\mu_{[x]}$ as $\varepsilon' \to 0$ and $N \to \infty$.*

**Interpretation.** Proposition 3.1 provides a practical method for consistently retrieving $\mu_{[x]}$ through *canonical orientation normalization*, outlined in practice in Algorithm 1. The process involves identifying the class $[x]$ and collecting the relative transformations $\psi([x])$ (representing poses relative to the arbitrary canonical $\hat{x}$). The distribution of transformation in this set is offset w.r.t. $\mu_{[x]}$ by a translation induced by the canonical pose, which can be estimated through the empirical Fréchet mean $\hat{\Gamma}_{[x]}$ of the observed transformations $\psi([x])$. Then, right-multiplying by the inverse $\hat{\Gamma}_{[x]}^{-1}$ corrects this offset and centers the Fréchet mean at the identity consistently for all classes. This removes the influence of the arbitrary choice of the canonical pose $\hat{x}$, providing a geometrically meaningful

---

**Algorithm 1** Canonical orientation normalization

**Require:** Trained IE-AE $(\eta, \delta, \psi)$, input $x$, number of neighbors $k$
1: Compute invariant embedding: $z \leftarrow \eta(x)$
2: Find $k$-nearest neighbors $s_j$ of $x$ based on $d_\mathcal{Z}(\eta(s_j), z)$ to approximate class $[x]$
3: Collect relative transformations:
$\psi([x]) \leftarrow \{\psi(s_j) \mid s_j \in \text{k-NN}(x)\}$
4: Estimate offset via Fréchet mean:
$\hat{\Gamma}_{[x]} \leftarrow \mathrm{argmin}_{y \in \mathcal{G}} \sum_{g_i \in \psi([x])} d_\mathcal{G}(y, g_i)^2$
5: Compute inverse offset: $\hat{\Gamma}_{[x]}^{-1}$
6: Compute normalized transformations:
$\psi'([x]) \leftarrow \{g_i \hat{\Gamma}_{[x]}^{-1} \mid g_i \in \psi([x])\}$
7: **return** $\psi'([x])$     ▷ Samples approximating $\mu_{[x]}$

---

canonical (centered at the Fréchet mean) and enabling the retrieval of symmetry distributions with useful properties, as we show in our experiments (Sec. 5).

### 3.3 INFERRING SYMMETRIES VIA LEARNED MAPPINGS

Algorithm 1 provides a way to estimate the symmetry distribution and the centering transformation for any class *in the training data*. To enable efficient inference for *unseen* inputs without explicit class computations at test time, we can train learnable mappings. If $\mu_{[x]}$ has a known parametric form (or we approximate it parametrically), its parameters $\theta_{[x]}$ can be estimated as

$$\hat{\theta}_{[x]} \;=\; \varphi\left(\psi\left([x]\right)\hat{\Gamma}_{[x]}^{-1}\right), \tag{4}$$

where $\varphi$ is an appropriate estimator for the parameters of the distribution, e.g., maximum likelihood. Consider the estimates $\hat{\theta}_{[x]}$ and $\hat{\Gamma}_{[x]}$. We can learn two maps using these estimates as pseudo-labels:

- A map $\Theta$ predicting the parameters of the symmetry distribution of an input by minimizing $\mathcal{L}_p = d_\theta(\Theta(x), \hat{\theta}_{[x]})$,
- A map $\Lambda : \mathcal{X} \to \mathcal{G}$, predicting the centering transformation by minimizing $\mathcal{L}_c = d_G(\Lambda(x), \hat{\Gamma}_{[x]})$,

where and $d_\theta, d_\mathcal{G}$ are appropriate distances. The first mapping generalizes the estimation process: at test time, given $x$, we can predict its symmetry parameters as $\hat{\theta} = \Theta(x)$ without requiring class computations. The second mapping allows us to obtain our RECON canonicalizations during inference as $C(x) = \rho_\mathcal{X}(\Lambda(x) \cdot \psi(x)^{-1})\, x$. Additionally, we can use these functions in combination to detect out-of-distribution symmetries Section 5.2. We model $\Theta$ and $\Lambda$ as $\mathcal{G}$-invariant networks, ensuring that predictions depend only on the object's class $[x]$, not its specific input pose.

## 4 RELATED WORK

**Class-pose decomposition methods**    Unsupervised learning of disentangled invariant and equivariant representations via autoencoders or other learning paradigms has seen various propositions (Shu et al., 2018; Guo et al., 2019; Feige, 2019; Kosiorek et al., 2019; Koneripalli et al., 2020; Winter et al., 2021; 2022; Yokota & Hontani, 2022). In this work, we build on top of the IE-AE framework (Winter et al., 2022). Our main contribution is the discovery of the transformations in the data via an invariant latent space search coupled with an explicit canonical orientation normalization step (Proposition 3.1), which corrects the pose offset introduced by arbitrary canonicals typical of class-pose decomposition methods like the IE-AE. In principle, however, this approach can be applied to any method which factors inputs into an invariant component and a symmetry component. For instance, Quotient Autoencoders (Yokota & Hontani, 2022) also learn canonical representations in a similar spirit; Marchetti et al. (Marchetti et al., 2023) proposes a class-pose decomposition network akin to IE-AEs, albeit with a different learning paradigm and advantages; SGM (Allingham et al., 2024) also feature a canonical representation or prototype and a relative transformation component. Our approach remains compatible with such backbones.

**Learning symmetries from data**    While standard group equivariant networks impose fixed symmetries (Cohen & Welling, 2016; Cohen et al., 2018; Weiler et al., 2018; Weiler & Cesa, 2019; Cohen et al., 2019; Romero et al., 2020; Romero & Cordonnier, 2020; Wang et al., 2020), recent effort have focused on learning symmetries from data. Approaches like Augerino (Benton et al., 2020) learn data augmentations for non-equivariant models, while others implement relaxed equivariance constraints. Partial G-CNNs modulate the equivariance per-layer by learning a distribution over the group (Romero & Lohit, 2022); Residual Pathway Priors (Finzi et al., 2021) propose handling partial equivariances through a combination of equivariant and non-equivariant models. These methods require supervision or learn dataset-level transformations rather than instance-specific distributions. Recently, Variational Partial Group Convolutions (VP G-CNNs)(Kim et al., 2024) extended Partial G-CNNs to adapt to instance-level symmetries with a variational inference approach. While they can compute a class-dependent "equivariance error", this method does not expose a clear symmetry distribution for each input. Equivariance via weight-sharing patterns (Ravanbakhsh et al., 2017; Zhou et al., 2021; Yeh et al., 2022) can also be leveraged to adapt to partial or dataset-level symmetries. Akin to Partial G-CNNs, WSCNNs (van der Linden et al., 2024) introduce layers that can adjust their equivariance based on the data by modulating the weight-sharing pattern. This approach learns a single set of transformations per layer, while our method discovers instance-level symmetry distributions instead.

Another line of work aims to learn a dataset-level symmetry (sub)group $\mathcal{H} \leq \mathcal{G}$ (or generators) of a prescribed ambience group $\mathcal{G}$ from data. LieGG (Moskalev et al., 2023) extracts infinitesimal generators from a trained model to reveal learned invariances; LieGAN (Yang et al., 2023) adversarially learns Lie algebra generators and a dataset-level distribution over coefficients on the group, discovering subgroups of a prescribed $\mathcal{G}$ without supervision, as well as subsets of the group through regularization strategies; LaLiGAN (Yang et al., 2024) lifts to a latent space to capture non-linear actions. Other methods reason through Lie derivatives (Otto et al., 2025) or extend to broader types of symmetries (Forestano et al., 2023; Shaw et al., 2024; Ko et al., 2024). In contrast to these methods, our approach discovers *instance-specific* distributions of symmetries over a group $\mathcal{G}$ and a data-aligned canonicalization function without supervision, rather than discovering a dataset-level (sub)group $\mathcal{G}$ itself. We emphasize that our method is complementary and a continuation to this line of work: these approaches can be used to infer a suitable $\mathcal{G}$ for all the data, while RECON aims to infer instance-level distributions and a test-time canonicalization operator.

Symmetry-aware Generative Model (SGM) (Allingham et al., 2024) is the most closely related method in terms of instance-level granularity. SGM uses a flow-based model to learn the relative distribution of transformations. However, this distribution is relative to an arbitrary canonical, which presents several disadvantages that RECON addresses (Figures 1, 2, 5b). Note that SGM is a class-pose decomposition method, and therefore is compatible with RECON. Preliminary work (Urbano & Romero, 2024) on class-pose methods proposed enforcing a constraint on the group action estimator via a regularization term, which can be leveraged to learn symmetries, but this often leads to degenerate solutions where $\psi(x) \approx e$ for all $x$, limiting symmetry learning. Other methods model probability distributions over the group. Implicit-PDF (Murphy et al., 2022) models instance-level distributions over $SO(3)$ with an implicit density and pose supervision; Alignist (Vutukur et al., 2024) estimates orientation distributions using CAD shape priors and correspondence distributions. In contrast, RECON requires neither pose labels nor CAD models and operates on general groups supported by the class-pose backbone.

**Test-time canonicalization.** Other works can grant model-agnostic group invariance to pre-trained models by inserting a small canonicalization in front of them. Spatial Transformer Networks (STN) (Jaderberg et al., 2015) learn an input-dependent geometric warp via a small network which can be seen as a group-free form of canonicalization, but it is typically trained end-to-end with a discriminative objective and requires joint training with the predictor. Equivariant Adaptation (EquiAdapt) (Mondal et al., 2023) places an equivariant canonicalizer before the frozen predictor and trains only the canonicalizer using a canonicalization prior; for smaller classifiers they also propose joint fine-tuning of the predictor. In both cases, training the canonicalization is tied to a specific pretrained model and its size. In contrast, our canonicalization is trained independently of the downstream model and can be plugged in front of any model that operates on the same input domain. In a similar fashion, Affine Steerable EquivarLayer (Li et al., 2025) learns a canonicalizer trained independently of the pre-trained model, but we note that it is trained using known random transforms, i.e. pose labels. Our canonicalization, however, is obtained without any supervision.

## 5 EXPERIMENTS

We empirically validate RECON on diverse datasets and rotational symmetry distributions. In particular, we use benchmark image datasets and a large-scale, real-world geometric graphs dataset exhibiting known ground truth $SO(2)$ and $SO(3)$ rotational symmetries respectively. We focus on rotation groups since rotations have been recognized as a significant challenge (specially in 3D) by several studies (Zhao et al., 2020; Shen et al., 2021) even for small ($<15°$) angles (Sun et al., 2022), but our framework remains general. All our results are on test data, using the learned $\Theta$ and $\Lambda$ predictors trained with the pseudo-labels obtained through Algorithm 1 (Sec. 3.3). Further experimental details are provided in Appx. E.

### 5.1 RECOVERING SYMMETRIES

**Imaging** We first validate RECON on rotated MNIST and FashionMNIST (Axelrod & Gómez-Bombarelli, 2022) datasets where different classes exhibit distinct rotational symmetry patterns. For MNIST, we apply random rotations drawn uniformly from $\pm 60°$ for digits 0-4 and $\pm 90°$ for digits 5-9. For FashionMNIST, we apply rotations drawn from a Gaussian $\mathcal{N}(0, \sigma)$ with $\sigma=0$ for classes 0-2, $\sigma=32$ for classes 3-5 and $\sigma=64$ for classes 6-9. We aim to discover these input-dependent symmetries from the unlabeled datasets. Canonicalization plots (Fig. 2) show RECON's consistent,

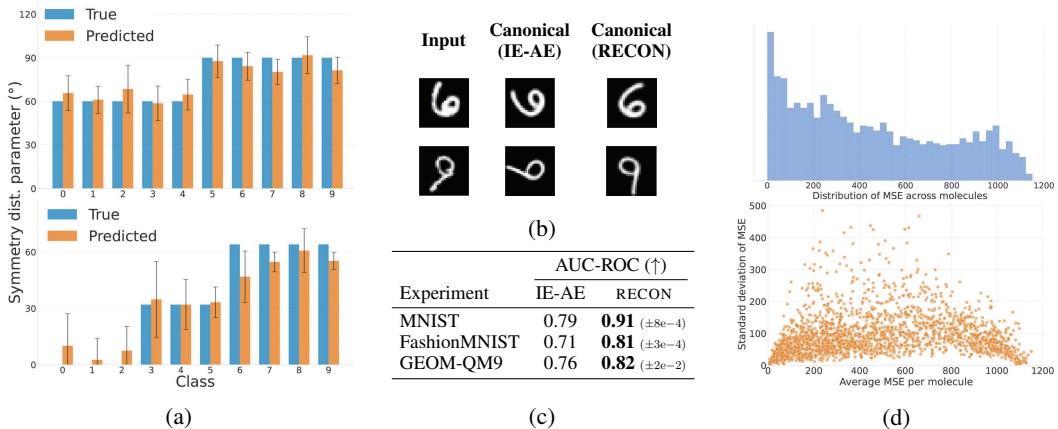

Figure 5: **(a)** True (■) vs predicted (■) parameter of the label-dependent symmetry distribution in MNIST (top) and FashionMNIST (bottom) experiment. **(b)** Canonicals for digits of the class 6 (first row) and 9 (second row) obtained with IE-AE and RECON. **(c)** Out-of-distribution detection performance with IE-AE and RECON. **(d)** Histogram of average prediction error across molecules and a scatter plot of their standard deviations for GEOM.

upright canonicals across classes, compared to other class-pose backbones. In Figure 9 Col.3, we visually confirm that our normalized distributions match the true sampling regime applied to each class in the MNIST experiment. Note how this clear distinction and direct interpretability is lost when analyzing the unnormalized, relative distributions from the class-pose backbone (Figure 9 Col.2). Figure 5a shows quantitative results for MNIST (top) and FashionMNIST (bottom), confirming that the predicted distribution parameters (given by $\Theta(x)$) align closely with the ground truth parameters, even when these symmetries stem from distributions with different shapes and scales per-class. We obtain these results using $k=10$ neighbors for the class computation $[x]$, which we found to give the best performance (details in App. D.2).

A notable strength of RECON is its ability to pick up on *partial symmetries*. RECON canonicalizations can distinguish between *distinct* classes that are related by a group transformation – such as digits '6' and '9', which are related by a $180°$ rotation – as opposed to the IE-AE canonicalizations, which collapse both classes into the same canonical[6] (Fig. 5b). This sensitivity to the data's contextual orientation allows RECON to handle cases beyond perfect group orbits where full equivariance is inappropriate. We discuss this in detail in Appendix D.4.

**Geometric graphs**    To demonstrate RECON's ability to discover symmetries in higher-dimensional groups and complex data, we apply it to the Geometric Ensemble of Molecules (GEOM) dataset (Axelrod & Gómez-Bombarelli, 2022), which provides multiple molecular conformations (i.e., unique 3D geometries or conformers) for several classes of molecules. We focus on the the QM9 (Ramakrishnan et al., 2014) subset of GEOM, and to ensure sufficient pose variation in each molecule class, select only molecules with at least 64 distinct low-energy conformers. This results in approximately 175k samples for training, 21k for validation, and 24k for testing across 2, 211 distinct classes of molecules, which can be identified by their SMILES (Anderson et al., 1987) string (more details in Appx. E). We then apply random $SO(3)$ rotations to each molecule class to create ground truth pose variations. Specifically, subsets of molecules are rotated using three distinct matrix-Fisher distributions (Mardia & Jupp, 2009), each centered on a different axis $e_1, e_2, e_3$ to simulate rotations around a fixed direction. These parameterized distributions $\{\mathcal{M}(F^1_{true}), \mathcal{M}(F^2_{true}), \mathcal{M}(F^3_{true})\}$ are visualized on the sphere following Mohlin et al. (2020), where brighter regions indicate preferred orientations. For instance, in Fig. 6a, rotations of `C[C@@]1(O)C[C@H]1[C@@H](O)CO` conformers are sampled from $\mathcal{M}(F^1_{true})$, which perturbs the $e_1$ axis locally while the orthogonal axes rotate around it freely forming a ring.

Figure 6 evaluates how accurately RECON recovers each molecule's $SO(3)$ symmetry distribution by measuring per-molecule average MSE between the true and predicted matrix-Fisher parameters. Panels (a-c) visualize randomly sampled molecules from three error regimes – low, average, and high

---

[6]This limitation can also be observed in SGM (Allingham et al., 2024).

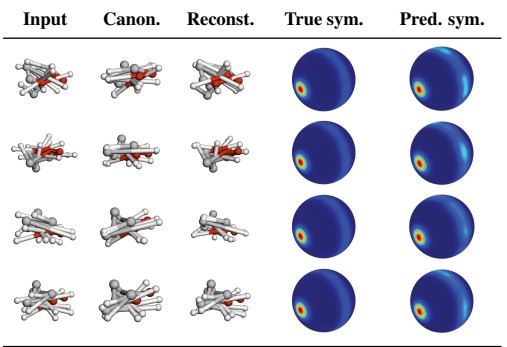

| Input | Canon. | Reconst. | True sym. | Pred. sym. |

(a) Low-error sample, MSE=173.71.

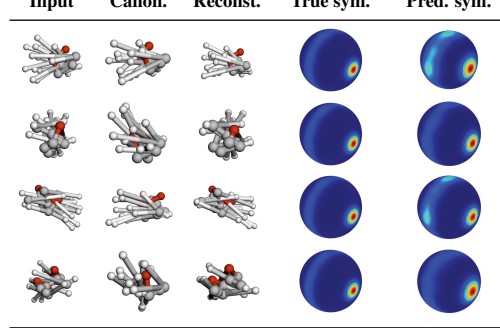

| Input | Canon. | Reconst. | True sym. | Pred. sym. |

(b) Average-error sample, MSE=368.91.

| Input | Canon. | Reconst. | True sym. | Pred. sym. |

(c) High-error sample, MSE=1074.76.

| Dataset (Train / Test) | Backbone | Canon. | Canon. Supervision | Test Acc. ($\uparrow$) |
|---|---|---|---|---|
| MNIST (Orig / Rot) | ResNet18 | - | - | $65.11 \pm 0.9\%$ |
| | | IE-AE | ✗ | $13.60 \pm 4.2\%$ |
| | | EquiAdapt | ✓ | $\mathbf{94.52} \pm 0.6\%$ |
| | | RECON | ✗ | $90.96 \pm 0.5\%$ |
| F-MNIST (Orig / Rot) | ResNet18 | - | - | $57.14 \pm 0.1\%$ |
| | | IE-AE | ✗ | $11.55 \pm 1.2\%$ |
| | | EquiAdapt | ✓ | $79.44 \pm 1.4\%$ |
| | | RECON | ✗ | $\mathbf{81.96} \pm 0.3\%$ |
| GEOM-QM9 (Orig / Rot) | GCN | - | - | $52.40 \pm 0.0\%$ |
| | | IE-AE | ✗ | $30.80 \pm 0.0\%$ |
| | | EquiAdapt | ✓ | $52.96 \pm 0.9\%$ |
| | | RECON | ✗ | $\mathbf{55.92} \pm 0.3\%$ |

(d) Test accuracies obtained using different test-time canonicalizations in pre-trained classifiers.

Figure 6: **(a-c)** Test-time comparison for molecules randomly sampled from low error ($\leq p_{10}$ tenth percentile average per-molecule MSE), average error ($p_{45} - p_{55}$), and high error ($\geq p_{90}$) regimes. For each selected molecule, four sample conformers are shown using a ball-and-stick model, displaying: (column 1) input conformer, (2) our canonical reconstruction, (3) final reconstruction, (4) true symmetry distribution (constant for the molecule), and (5) per-conformer predicted symmetry distribution. **(d)** Test-time canonicalization experiment on the per-class rotated dataset variants.

MSE based on percentiles. The panel visualizations show close alignment between the predicted and the true distributions for molecules in the low and average error regime. Figure 5d provides a global view, showing the histogram of MSE and their standard deviations. In general, about 80% of molecules fall within a region of accurate predictions (approx. MSE $< 800$ based on visualizations), demonstrating RECON's ability to capture diverse $SO(3)$ symmetry patterns from large-scale, unlabeled, realistic 3D data. Interestingly, we observe no clear correlation between MSE and (i) molecular size, (ii) flexibility in conformational variation or (iii) reconstruction error (Figure 14, reconstruction losses defined in Appendix E.2.4). This indicates that our symmetry estimates are robust to conformational complexity, and that the remaining error stems from other sources (see limitations in Sec. 6). RECON also demonstrates remarkable data efficiency, providing accurate symmetry distributions inferences in a dataset where most molecule classes have just a few (<100) number of conformers per class (Fig 13b), highlighting its ability to work with limited per-class data.

## 5.2 DOWNSTREAM APPLICATIONS

**OOD pose detection** The centered distributions recovered by RECON yield a natural anomaly score. Given predictors $\Theta(x)$ and $\Lambda(x)$ (Sec. 3.3), consider the absolute pose $g_{\text{abs}} = \psi(x)\Lambda(x)^{-1}$. Then, the likelihood of $g_{abs}$ under the distribution parameterized by $\Theta(x)$ serves as an anomaly score:

$$s(x) = -\log p_{\Theta(x)}(g_{\text{abs}}). \tag{5}$$

Low $s(x)$ indicates in-distribution, while high $s(x)$ flags OOD. We empirically validate this by classifying randomly oriented $SO(2)$ (images) and $SO(3)$ (GEOM) test instances, showing strong OOD detection (AUC-ROC in Table 5, measuring separability between in and OOD predictions).

While this score is reference-frame invariant in theory (we provide a proof in Prop. B.2), centering via RECON improves optimization, resulting in consistently higher AUC-ROC scores against IE-AE (full ablation details in Appendix F.1, ROC curves in Fig. 15).

**Test-time canonicalization**    RECON's data-aligned canonicalizations can be used to create a drop-in layer that grants group invariance to frozen backbones (e.g., classifiers, foundation models) at inference, *with no retraining*. Concretely, we canonicalize inputs at test time via a canonicalization layer defined as $C(x) = \rho_{\mathcal{X}}(\Lambda(x) \cdot \psi(x)^{-1}) x$, and feed the canonicalized sample to the pre-trained model; because RECON aligns inputs to the training data distribution of poses (visualized in Fig. 1d), it recovers a large fraction of the accuracy lost to test-time transformations.

Figure 6d shows considerable performance gains in classifiers. We improve consistently over the baseline and offer competitive (MNIST) or better performance (FashionMNIST, GEOM-QM9) against other test-time canonicalization methods such as EquiAdapt (Mondal et al., 2023), while noting that our canonicalization is trained without supervision, whereas EquiAdapt's canonicalization is trained end-to-end using class labels. Notably, in the imaging domain, our method consistently outperforms equivariance-learning architectures ($SE(2)$ Partial G-CNNs (Romero & Lohit, 2022)) sometimes by a large margin (67.72% vs 81.96% in FashionMNIST), and stay competitive (MNIST) or even surpass (FashionMNIST) fully equivariant architectures ($SE(2)$-equivariant Steerable CNNs (Cesa et al., 2022)) (cf. Table 3). Note that arbitrary canonicalizations from class-pose methods (e.g., IE-AE, SGM) map inputs to OOD poses and therefore degrade performance; benefits from these canonicalizations arise only when they are also applied at training time, thus requiring backbone retraining. Frame-averaging approaches face the same practical barrier, typically requiring incorporating the frame-averaging into training to grant invariance (Puny et al., 2022; Kaba et al., 2023), which limits adoption due to the added training cost. Our canonicalizations work as a plug-in solution during inference, facilitating its broader adoption. For further details, refer to Appendix F.2.

## 6    LIMITATIONS AND FUTURE WORK

We addressed the challenge of learning instance-specific symmetries from unlabeled data by leveraging class-pose decompositions and solving the problem of arbitrary canonicals. In our testing, RECON proved to be successful in recovering pose distributions at the instance-level – including demonstrations in 3D data. We also showed how RECON enables downstream practical applications, notably in test-time canonicalization, which yields clear performance gains and has potential for broader adoption due to its plug-and-play, architecture agnostic, retrain-free nature.

**Limitations**    A crucial limitation is that RECON relies on class-pose decompositions, where the relative transformation is defined with respect to the entire input $x \in \mathcal{X}$. This is effective in certain domains, but when we use class-pose methods as backbones for symmetry discovery e.g. in natural images, it exposes a key mismatch: the group action $\rho(g)$ moves the whole scene (input), whereas the object of interest occupies only a subset (e.g., a car within a street scene). Background and multi-object context make unsupervised association across instances difficult, hindering recovery of object-level symmetry distributions. A natural next step is moving from instance-level decomposition to an *object-centric* variant that predicts pose from token or mask-based features from e.g. strong self-supervised backbones. This preserves our pipeline's simplicity while targeting in-the-wild settings where symmetries are local to objects rather than inputs. Complementary improvements target sampling and modeling; for instance, replacing nearest neighbors with more sophisticated sampling strategies. On the density side, more flexible estimators for the symmetry distribution – like flow-based models in the spirit of Allingham et al. (2024), can better capture complex symmetry patterns than a parametric family. Finally, relaxing the need to predefine $\mathcal{G}$ by inferring group structure or group-agnostic transformations (van der Linden et al., 2024) is a promising direction. These extensions preserve RECON's simplicity while pushing toward more robust performance.

ACKNOWLEDGMENTS

We thank Moritz Wagner for valuable discussions and feedback on earlier drafts of the manuscript, and Carlos Saavedra Luque for his contributions to the design of the main figure. This research was partially supported by the Deutsche Forschungsgemeinschaft (DFG) through the DFG Cluster of Excellence MATH+ (EXC-2046/1, EXC-2046/2, project id 390685689), as well as by the German Federal Ministry of Research, Technology and Space (research campus Modal, fund number 05M14ZAM, 05M20ZBM) and the VDI/VDE Innovation + Technik GmbH (fund number 16IS23025B).

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

## A    BACKGROUND

**Groups and group actions.** A group $\mathcal{G}$ is a set equipped with a closed, associative binary operation $\cdot$ such that $\mathcal{G}$ contains an identity element $e \in G$ and every element $g \in \mathcal{G}$ has an inverse $g^{-1} \in \mathcal{G}$. For a given set $\mathcal{X}$ and group $\mathcal{G}$, the (left) group action of $\mathcal{G}$ on $\mathcal{X}$ is a map $\rho : \mathcal{G} \times \mathcal{X} \to \mathcal{X}$ that preserves the group structure. Intuitively, it describes how set elements transform by group elements.

**Group representations.** In this work, we focus on cases where $\mathcal{X}$ is a vector space. In such scenarios, the group acts on it by means of *group representations*. A representation of a group $\mathcal{G}$ on a vector space $\mathcal{X}$ is a homomorphism $\rho_{\mathcal{X}} : \mathcal{G} \to \mathrm{GL}(\mathcal{X})$ mapping each $g \in \mathcal{G}$ to an invertible linear operator on $\mathcal{X}$. We consider our dataset to be of the form $\mathcal{X} = \{f \mid f : \mathcal{V} \to \mathcal{W}\}$, where $\mathcal{V}$ is a set on which $\mathcal{G}$ acts, and $\mathcal{W}$ is a vector space on which $\mathcal{G}$ may also act. As an example, molecular conformations can be interpreted as functions $f : \mathcal{V} \to \mathbb{R}^3$ that map atoms to their spatial coordinates. Following this definition, a group element acts in a data sample as

$$[\rho_{\mathcal{X}}(g)f](x) \equiv \rho_{\mathcal{W}}(g) f \left( \rho_{\mathcal{V}}(g^{-1})x \right) . \tag{6}$$

For instance, a rigid transformation $g$ might shift or rotate node coordinates via $\rho_{\mathcal{W}}(g)$ while leaving the nodes unchanged or transforming them via $\rho_{\mathcal{V}}(g)$. Whenever we speak of a representation $\rho_{\mathcal{X}}$ on $\mathcal{X}$, it is understood that we are implicitly referring to the previous equation to understand the transformation of each component.

**Orbits.** The orbit of $x$, $\mathcal{O}_x = \{\rho_{\mathcal{X}}(g)x\}_{g \in \mathcal{G}}$ captures all possible transformations of $x$ resulting from the action of all elements of $\mathcal{G}$.

**Equivalence classes and quotient sets.** Our analysis relies on the definition of equivalence classes and their quotient sets. Let $\sim$ be an equivalence relation on $\mathcal{X}$ and consider the *equivalence classes* $[x] = \{y \in \mathcal{X}, \text{ s.t. } x \sim y\}$ of $\mathcal{X}$. The quotient set $\mathcal{X}/\sim$ is defined as the collection of all equivalent classes in $\mathcal{X}$ under the relation $\sim$.

**Group equivariance and group invariance.** A map $h : \mathcal{V} \to \mathcal{W}$ is $\mathcal{G}$-equivariant with respect to the representations $\rho_{\mathcal{V}}, \rho_{\mathcal{W}}$ if $h(\rho_{\mathcal{V}}(g)x) = \rho_{\mathcal{W}}(g)h(x) \; \forall g \in \mathcal{G}, \forall x \in \mathcal{X}$. In the context of neural networks, G-CNNs (Cohen & Welling, 2016) are designed to be $\mathcal{G}$-equivariant by using only $\mathcal{G}$-equivariant layers in their constructions. This ensures that applying a transformation $g \in \mathcal{G}$ before a layer yields an equivalently transformed output. Analogously, a map $h$ is $\mathcal{G}$-invariant with respect to $\rho_{\mathcal{V}}$ if $h(\rho_{\mathcal{V}}(g)x) = h(x) \; \forall g \in \mathcal{G}, \forall x \in \mathcal{X}$. That is, if $\mathcal{G}$-transformations of the input yield the same result.

## B    PROOFS

**Proposition B.1** (Approximation of $\mu_{[x]}$ via Normalization). *Let $\mathcal{X}$ be a metric space, $\mathcal{G}$ a Lie group and $\eta, \delta, \psi$ an IE-AE where $\psi$ is continuous on a compact domain $\mathcal{X}$. Suppose that $\mathcal{X}$ exhibits symmetries characterized by $\mu_{[x]}$ where $\mathcal{F}(\mu_{[x]}) = e$ as described above. Consider a random sample $\{s_i\}_{i=1}^N$ of $[x]$ and denote their images by $\psi$ as $\psi([x]) = \{\psi(s_i)\}_{i=1}^N$. Let $\hat{\Gamma}_{[x]} \in \mathcal{G}$ be the empirical Fréchet mean of $\psi([x])$. Then, the empirical distribution $\hat{\mu}_{[x]}$ corresponding to the normalized samples $\psi([x])\hat{\Gamma}_{[x]}^{-1}$ approximates the target distribution $\mu_{[x]}$. Specifically, $\hat{\mu}_{[x]}$ converges in Wasserstein distance to $\mu_{[x]}$ as $\varepsilon' \to 0$ and $N \to \infty$.*

*Proof.* Consider $N$ independent samples $\{s_i\}_{i=1}^N$ drawn from class $[x]$ according to the model $s_i = \rho_{\mathcal{X}}(g_i)\gamma_{[x]} + \varepsilon_i$, where $g_i \sim \mu_{[x]}$ i.i.d. and $\varepsilon_i \sim P_\epsilon$ i.i.d, with $||\varepsilon_i|| < \varepsilon'$ almost surely. Let $\psi([x]) = \{\psi(s_i)\}_{i=1}^N$ be the set of observed relative transformations and $\nu_{[x]}$ be the distribution over $\mathcal{G}$ from which $\psi(s_i)$ are sampled. Our first goal is to show that $\nu_{[x]}$ and $\mu_{[x]}$ differ by some translation $\Gamma_{[x]} \in \mathcal{G}$.

Let $s_i' = \rho_{\mathcal{X}}(g_i)\gamma_{[x]}$ be the noise-free (i.e., $\varepsilon_i = 0$) counterparts of $s_i$, and consider their images $\psi(s_i')$. Denote $\nu_{[x]}'$ as the corresponding distribution over $\mathcal{G}$ from which $\psi(s_i')$ are sampled. Then,

$$\psi(s_i') \underset{\psi \;\; \mathcal{G}-\mathrm{eq}}{=} g_i \psi(\gamma_{[x]}) = g_i \Gamma_{[x]} \tag{7}$$

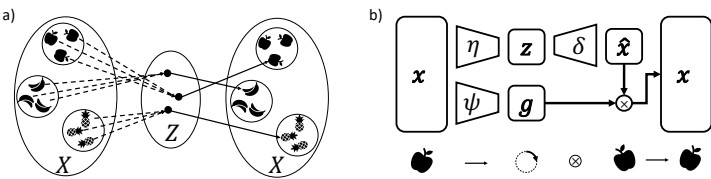

Figure 7: IE-AE architecture visualization. Image taken from (Winter et al., 2022).

for $\Gamma_{[x]} := \psi(\gamma_{[x]}) \in \mathcal{G}$. Since $g_i \sim \mu_{[x]}$, the distribution $\nu'_{[x]}$ is exactly the right-translate $\mu_{[x]}\Gamma_{[x]}$. That is, $\nu'_{[x]} = \mu_{[x]}\Gamma_{[x]}$. This relates the noise-free $\nu'_{[x]}$ and $\mu_{[x]}$. Now, we will show that this relation also holds approximately for $\nu_{[x]}$.

By the uniform continuity of $\psi$ (which holds since $\mathcal{X}$ is compact and $\psi$ is continuous), there exists a modulus of continuity $\omega$ such that $\lim_{\delta \to 0} \omega(\delta) = 0$ and for $s_i, s'_i \in \mathcal{X}$, it holds:

$$d_{\mathcal{G}}(\psi(s_i), \psi(s'_i)) \leq \omega(d_{\mathcal{X}}(s_i, s'_i)) = \omega(||\varepsilon_i||) < \omega(\varepsilon') = K, \tag{8}$$

for some small $K > 0$, since $\varepsilon'$ is presumably small and $\lim_{\delta \to 0} \omega(\delta) = 0$. This establishes pointwise closeness almost surely.

Consider the $p$-Wasserstein distance between both probability distributions, defined as

$$W_p(\nu_{[x]}, \nu'_{[x]}) = \inf_{\pi \in \Pi(\nu_{[x]}, \nu'_{[x]})} \left( \int_{\mathcal{G} \times \mathcal{G}} d_{\mathcal{G}}(g_1, g_2)^p d\pi(g_1, g_2) \right)^{1/p}, \tag{9}$$

where $\Pi(\nu_{[x]}, \nu'_{[x]})$ is the set of all possible couplings (joint probability measures) on $\mathcal{G} \times \mathcal{G}$ with marginals $\nu_{[x]}$ and $\nu'_{[x]}$. Consider the natural coupling $\pi_{nat}$ determined by the joint random variable pair $(\psi(s_i), \psi(s'_i))$. $\pi_{nat}$ is a valid coupling in $\Pi(\nu_{[x]}, \nu'_{[x]})$, since its marginals are $\nu_{[x]}$ and $\nu'_{[x]}$ respectively. Since the Wasserstein metric is defined as the minimum over all possible couplings, it must hold that

$$W_p(\nu_{[x]}, \nu'_{[x]}) \leq \left( \int_{\mathcal{G} \times \mathcal{G}} d_{\mathcal{G}}(g_1, g_2)^p \, d\pi_{nat}(g_1, g_2) \right)^{1/p} = \tag{10}$$

$$= \left( \int_{\mathcal{G} \times \mathcal{G}} d_{\mathcal{G}}(\psi(s_i), \psi(s'_i))^p \, d\pi_{nat}(g_1, g_2) \right)^{1/p} < \left( \int_{\mathcal{G} \times \mathcal{G}} \omega(\varepsilon')^p \, d\pi_{nat}(g_1, g_2) \right)^{1/p} = \omega(\varepsilon'). \tag{11}$$

This inequality shows that the distribution $\nu_{[x]}$ generating our samples $\psi([x])$ is close to the distribution $\nu'_{[x]} = \mu_{[x]}\Gamma_{[x]}$ in Wasserstein distance, with proximity controlled by the noise bound $\varepsilon'$ via the modulus of continuity $\omega$. Let's show now how to estimate the unknown $\Gamma_{[x]}$.

First, note that we can calculate the Fréchet mean of $\nu'_{[x]}$ as follows:

$$\mathcal{F}(\nu'_{[x]}) = \mathcal{F}(\mu_{[x]})\Gamma_{[x]} = e\Gamma_{[x]} = \Gamma_{[x]}, \tag{12}$$

where we used the property that the Fréchet mean is equivariant under isometries (such as right-translation by $\Gamma_{[x]}$ when using a right-invariant metric), $\mathcal{F}(\mu\Gamma) = \mathcal{F}(\mu)\Gamma$ (Karcher, 1977). This means that the unknown $\Gamma_{[x]}$ is the Fréchet mean of $\nu'_{[x]}$. However, we just proved that $\nu'_{[x]}$ and $\nu_{[x]}$ are close, bounded by $\omega(\varepsilon')$ (equation 10). We want to prove now that their Fréchet means are also close.

Consider the Fréchet mean $\mathcal{F}(\nu_{[x]})$ of the actual data distribution. The Fréchet functional is known to be strictly convex within geodesic balls of a certain radius $r_0$ determined by the manifold's geometry (Karcher, 1977). Within such regions, the Fréchet mean map $\mu \mapsto \mathcal{F}(\mu)$ is Lipschitz continuous with respect to the Wasserstein distance (Afsari, 2011). Therefore, exists a constant $C_L > 0$ such that

$$d_{\mathcal{G}}(\mathcal{F}(\nu_{[x]}), \mathcal{F}(\nu'_{[x]})) \leq C_L W_p(\nu_{[x]}, \nu'_{[x]}) < C_L \omega(\varepsilon'). \tag{13}$$

Substituting $\mathcal{F}(\nu'_{[x]}) = \Gamma_{[x]}$, we have $d_{\mathcal{G}}(\mathcal{F}(\nu_{[x]}), \Gamma_{[x]}) < C_L \omega(\varepsilon')$. This shows the true Fréchet mean of the distribution we sample from is close to $\Gamma_{[x]}$. Therefore, we can estimate $\Gamma_{[x]}$ by obtaining the population Fréchet mean of $\nu_{[x]}$.

Let's show that using the sample Fréchet mean of $\psi([x])$ estimates the true Fréchet mean. In effect, the sample Fréchet mean is a statistically consistent estimator of the population Fréchet mean $\mathcal{F}(\nu_{[x]})$ (Aveni & Mukherjee, 2024):

$$\hat{\Gamma}_{[x]} \xrightarrow{P} \mathcal{F}(\nu_{[x]}) \quad \text{as } N \to \infty. \tag{14}$$

Combining all the above, we see that for large $N$ and small $\varepsilon'$, $\hat{\Gamma}_{[x]}$ provides a good approximation of $\Gamma_{[x]}$.

Finally, consider the Fréchet-normalized empirical measure $\hat{\mu} = \frac{1}{N} \sum_{i=1}^{N} \delta_{\psi(s_i)\hat{\Gamma}_{[x]}^{-1}}$. By the continuous mapping theorem, and because group inversion and multiplication are continuous operations in the Lie group, it follows that $\hat{\mu}$ converges to $\mu_{[x]}$ in $W_p$ distance. $\square$

**Proposition B.2** (Right-translation invariance of the log density). *Let $\mathcal{G}$ be a Lie group with a right Haar measure $\lambda$. Let $\mu$ be a probability measure on $\mathcal{G}$ absolutely continuous w.r.t. $\lambda$, with density $p_\mu = \frac{d\mu}{d\lambda} \in L^1(\lambda)$. Fix $\gamma \in \mathcal{G}$ and let $r_\gamma(g) = g\gamma$. If $\nu = (r_\gamma)_* \mu$, then*

$$p_\nu(h) = p_\mu(h\gamma^{-1}) \quad \text{for almost every } h \in \mathcal{G}. \tag{15}$$

*Consequently,*

$$-\log p_\nu(h) = -\log p_\mu(h\gamma^{-1}) \quad \text{for almost every } h \in \mathcal{G}, \tag{16}$$

*interpreted in the extended reals with the convention $-\log 0 = +\infty$.*

*Proof.* For any bounded measurable $f : \mathcal{G} \to \mathbb{R}$,

$$\int f(h) \, d\nu(h) = \int f(r_\gamma(g)) \, d\mu(g) = \int f(r_\gamma(g)) \, p_\mu(g) \, d\lambda(g). \tag{17}$$

Set $h = g\gamma$. Right invariance of $\lambda$ gives $d\lambda(h) = d\lambda(g)$, hence

$$\int f(h) \, d\nu(h) = \int f(h) \, p_\mu(h\gamma^{-1}) \, d\lambda(h). \tag{18}$$

By uniqueness of Radon-Nikodym derivatives, $p_\nu(h) = p_\mu(h\gamma^{-1})$ for almost every $h$. Taking $-\log$ yields equation 16 (with $-\log 0 = +\infty$). $\square$

## C  ROBUST FRÉCHET MEAN EXTENSION FOR MULTIMODAL DISTRIBUTIONS

While the Fréchet mean provides a robust and well-defined centroid for many distributions, as discussed above, it may fall outside the support in certain multimodal cases, leading to non-natural canonical poses that do not align with the data's intrinsic symmetries. For instance, consider a bimodal distribution on $SO(2)$ with identical modes at 0 and $\pi$ radians. Here, the Fréchet mean minimizes the sum of squared geodesic distances and, due to the symmetry, converges to a point midway between the modes ($\pm\pi/2$), which lies in a region of near-zero probability density, outside the effective support of the distribution. This violates the desired natural (in-distribution) reference pose, as it may correspond to an orientation not observed in the data, potentially degrading downstream applications like invariance granting or interpretability.

To address this limitation while preserving the generality of the Fréchet mean, we propose an extension using *redescending M-estimators* (Shevlyakov et al., 2008; Rousseeuw & Hubert, 2011). These estimators replace the squared loss with a bounded, redescending function that caps the influence of distant points, providing robustness to outliers or separated modes. We specifically propose using the Tukey biweight loss (Rousseeuw & Hubert, 2011; Shin & Oh, 2022):

$$m(u; c) = \begin{cases} \frac{c^2}{6} \left( 1 - \left( 1 - \left(\frac{u}{c}\right)^2 \right)^3 \right) & \text{if } u \leq c \\ \frac{c^2}{6} & \text{if } u > c \end{cases} \tag{19}$$

In the standard Fréchet mean, we minimize $\sum_i d(y, g_i)^2$, which is unbounded quadratic in distances; large distances (outliers or far modes) have disproportionate pull because their squared penalty grows without limit. However, we can provide a robust version by minimizing $\sum_i m(d(y, g_i); c)$ instead.

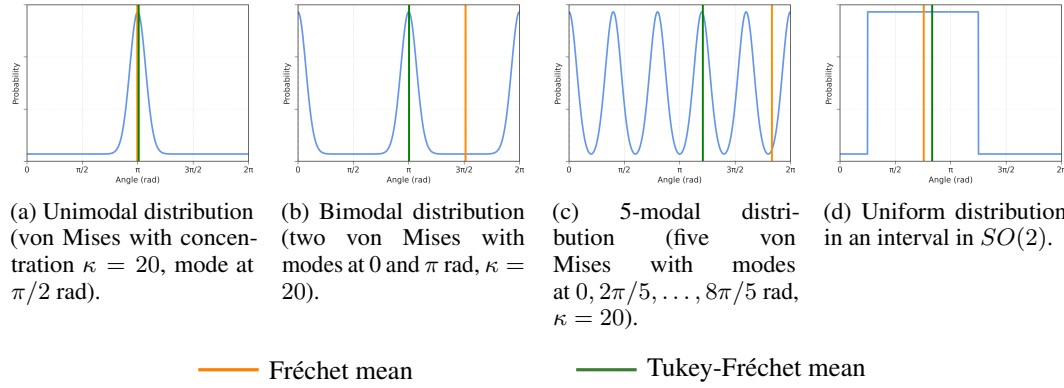

(a) Unimodal distribution (von Mises with concentration $\kappa = 20$, mode at $\pi/2$ rad).

(b) Bimodal distribution (two von Mises with modes at 0 and $\pi$ rad, $\kappa = 20$).

(c) 5-modal distribution (five von Mises with modes at $0, 2\pi/5, \ldots, 8\pi/5$ rad, $\kappa = 20$).

(d) Uniform distribution in an interval in $SO(2)$.

——— Fréchet mean            ——— Tukey-Fréchet mean

Figure 8: Comparison of Fréchet mean and robust Tukey-Fréchet mean variation on $SO(2)$ distributions. Plots generated by sampling $n = 2000$ points from each distribution, computing the means, and overlaying on the theoretical density. The robust estimator localizes to high-density regions in multimodal cases, while converging to the standard Fréchet mean in unimodal/uniform scenarios.

In effect, the robustness arises from the *bounded influence* of the Tukey biweight loss; $m$ is such that for small $u$, it holds that $m(u) \approx u^2/2$, behaving like the Fréchet mean locally (quadratic). However, for large distances $u > c$, $m$ is constant at $c^2/6$, assigning zero gradient to distant points:

$$\frac{dm}{du}(u; c) = \begin{cases} u\left(1 - \left(\frac{u}{c}\right)^2\right)^2 & \text{if } |u| \leq c \\ 0 & \text{if } |u| > c \end{cases} \tag{20}$$

The function $dm/du$, determines how much each sample affects the final estimate during optimization (we can use gradient descent to solve the minimization problem as in Shin & Oh (2022)). Therefore, in a multimodal distribution with well-separated clusters, large distances ($|u| > c$) have zero influence on the gradient/update, and the objective effectively ignores points from secondary modes when evaluated near a primary mode, minimizing only over the local cluster. To mitigate ambiguity, initialization at a common starting point can guide convergence to a "principal" consistent mode. In contrast, the Fréchet mean's quadratic penalty ($u^2$) grows unbounded, pulling the minimizer toward a global compromise, often outside any cluster. We provide some simulations in $SO(2)$ in Fig. 8.

The parameter $c > 0$ controls the robustness threshold (e.g., $c = \pi/4$ for $SO(2)$, half the maximum geodesic distance $\pi$; more generally, $c$ can be set adaptively, but intuitively, smaller values of $c$ induce a "shorter-sight" on the estimator).

Formally, given a distribution $\mu$ over the Lie group $\mathcal{G}$ with geodesic distance $d_\mathcal{R}$, the robust centroid or *Tukey-Fréchet mean* $\mathcal{F}_r(\mu)$ is defined as:

$$\mathcal{F}_r(\mu) = \arg\min_{y \in \mathcal{G}} \mathbb{E}_{g \sim \mu}[m(d_\mathcal{R}(y, g); c)], \tag{21}$$

where $m(u; c)$ is the Tukey biweight loss. For empirical samples $\{g_i\}_{i=1}^n$, the estimator becomes:

$$\hat{\mathcal{F}}_r = \arg\min_{y \in \mathcal{G}} \sum_{i=1}^n m(d_\mathcal{R}(y, g_i); c). \tag{22}$$

## C.1 EXAMPLES

We show comparison via simulations between the Fréchet mean and the robust variation for common distributions (Fig. 8). In $SO(2)$, the Fréchet mean has a closed-form expression (circular mean). The robust variation lacks a closed form and requires optimization, e.g., scalar minimization over $[0, 2\pi]$), but as a one-time computation for pseudo-labels (Algorithm 1), this is feasible (e.g., <1s for $n = 2000$ on standard hardware). In multimodal cases, it converges to a mode (e.g., $\pi$ for bimodal, as shown in 8b) while Fréchet converges to $\pm\pi/2$. In $SO(3)$, there is no closed form for either Fréchet mean or the robust variation. In this case, the Fréchet mean can be found via singular value decomposition or Riemannian gradient descent.

# D    ADDITIONAL INSIGHTS AND EXPERIMENTS DETAILS

## D.1    ORBITS, STABILIZERS, AND RELATION TO INSTANCE-LEVEL POSE DISTRIBUTIONS

We clarify how our notion of instance-level pose/orbit distributions relates to the group-theoretic notions of orbits and stabilizers. Given $x \in \mathcal{X}$ and a group $\mathcal{G}$ acting on $\mathcal{X}$, the stabilizer $\mathcal{S}_x = \{g \in \mathcal{G} \text{ s.t. } gx = x\}$ collects the (self) symmetries of $x$. For example, on planar rotations, a perfect circle has $\mathcal{S}_{\text{circle}} = SO(2)$, whereas a perfect square has $\mathcal{S}_{\text{square}}$ equal to the group of $90°$ rotations.

In contrast, our instance-level distributions $\mu_{[x]}$ do not model stabilizers explicitly; instead, they approximate the part of the orbit $\mathcal{O}_x$ that is actually observed in the data. When $x$ has a non-trivial stabilizer $\mathcal{S}_x$, many group elements act identically on $x$, so the pose distribution is only identifiable modulo $\mathcal{S}_x$ and becomes broad or multimodal along the corresponding symmetry directions. This makes canonical poses inherently ambiguous for highly symmetric inputs, but our Tukey-Fréchet estimator is explicitly designed to remain robust in this regime.

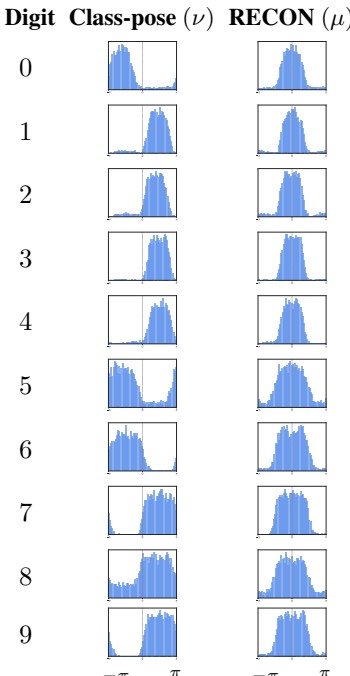

Figure 9: Probability density of recovered $SO(2)$ distributions on MNIST experiment: IE-AE, vs RECON.

This is visible for the digit "8" in the per-class rotated MNIST experiment: for a small fraction of very symmetric handwritten 8s, upright and upside down configurations are effectively indistinguishable, so the learned pose distribution shows a non-negligible mass near $180°$ (pose ambiguity), unlike digits such as 3, 4, or 7 whose histograms are sharply concentrated. In such cases, our Tukey-Fréchet objective does not average the modes into an unstable in-between pose; it admits minimizers aligned with one of the symmetric modes, yielding stable in-distribution canonicalizations (cf. Appendix C) and enabling test-time canonicalization.

## D.2    IMPACT OF NEIGHBORHOOD SIZE

Our symmetry estimation relies on approximating the class $[x]$ via $k$-nearest neighbors in the invariant space $\mathcal{Z}$. Figure 10 shows the impact of $k$ on the MAE of the predicted per-class symmetry parameter $\theta$ for the MNIST experiment. While Proposition 3.1 suggests convergence as $N \to \infty$ (larger $k$), practical datasets have finite class separation and noisy samples. Large $k$ can include samples from different underlying classes in the $k$-NN approximation, increasing noise and impacting performance. Conversely, very small $k$ may not provide a representative sample. Overall, $k = 25$ offers the best balance for GEOM-QM9 and $k = 10$ for MNIST and FashionMNIST, and the performance on the symmetry discovery task is not overly sensitive to the neighborhood size within a reasonable range of neighbors, indicating that our method is stable w.r.t. the choice of $k$.

## D.3    ANALYSIS OF EQUIVALENCE CLASS DEFINITION

To quantitatively assess the quality of the computed equivalence classes $[x]$, we introduce a *hit rate* metric that, for each $x \in \mathcal{X}$, measures the proportion of $k$-nearest neighbors in the computation of $[x]$ that belong to the same molecular class (SMILES) as $x$. A high hit rate indicates that our definition of equivalence class captures pose-invariant similarity successfully.

Figure 13a shows the distribution of hit rates across all $174,481$ conformers in our GEOM-QM9 dataset. The average hit rate is $0.95$, with $76.8\%$ of distinct molecules achieving an average hit rate above $0.9$. This confirms that the proposed equivalence class definition behaves as expected, abstracting molecular identity across conformational variations and group transformations, while maintaining discriminative power between different molecular structures. Figure 13b shows that molecules with more conformers in the dataset tend to achieve slightly higher hit rates (correlation

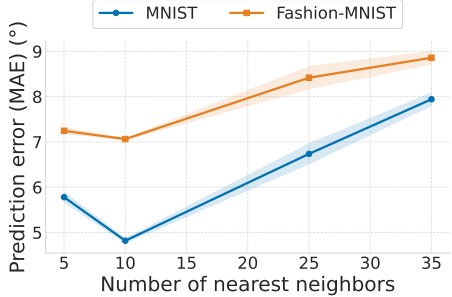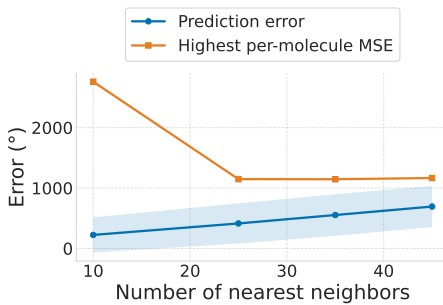

Figure 10: Prediction error in symmetry discovery task vs number of neighbors in class computation for MNIST and FashionMNIST experiments (left) and GEOM-QM9 (right). On the left, we plot average prediction error, which is defined as the MAE (in degrees) between the ground truth symmetry parameter and the per-class average predicted parameter, averaged across classes; shaded bands show the corresponding standard deviation across multiple runs with different seeds. On the right, we plot average and standard deviation of the prediction error, which is defined by the MSE between predicted and true matrix-Fisher parameters, averaged over all conformers. The orange curve reports the highest per-molecule mean MSE across conformers, illustrating the trade-off between average accuracy and worst case error as the number of neighbors varies.

coefficient $0.149$). This is expected; molecules with a small number of conformers are more likely to include neighbors that belong to other molecules. This supports the insights about how the number of neighbors affects the quality of the symmetry discovery (Fig. 10), since a greater number of neighbors increases the probability of a reduced hit rate on low-conformer molecules.

We also report hit rate metrics for varying number of neighbors on the imaging experiments in Fig. 13c. Note that configurations with the highest hit rate ($k = 5$) do not necessarily correspond to configurations with lowest prediction error ($k = 10$) in the symmetry discovery task (Fig. 10, left). In effect, lower $k$ generally yields higher hit rates, but using too few neighbors provides insufficient samples for accurate estimation. This explains the observed trade-off in Fig. 10, where $k = 10$ achieves the lowest prediction error despite $k = 5$ having the highest hit rate.

**Limitations of the proposed equivalence class**
We discuss the most prominent failure modes of our equivalence class definition by examining lowest-hit-rate cases in FashionMNIST, the dataset which exhibited the lowest hit rate score in our experiments (cf. Figure 13c). The heatmap in Figure 11 shows that most inputs with equivalence class construction imprecisions concentrate on a few semantically similar labels, particularly the footwear block (*Sneaker* $\leftrightarrow$ *Ankle boot*) and tops (*Coat* $\leftrightarrow$ *Shirt*). For a qualitative analysis, we visualize at random some equivalence classes with low and zero hit rate in Figure 12. We observe that neighbors of each input are visually nearly indistinguishable from the input, up to small cues (e.g., ankle height), partly due to small resolution of the data. As a result, the encoder maps them together in the latent space, despite having different semantic labels, which generates the feature overlap.

Overall, these figures indicate that the most challenging scenarios for our method involve inputs with *high interclass overlap in the latent space* (and not large intra-class

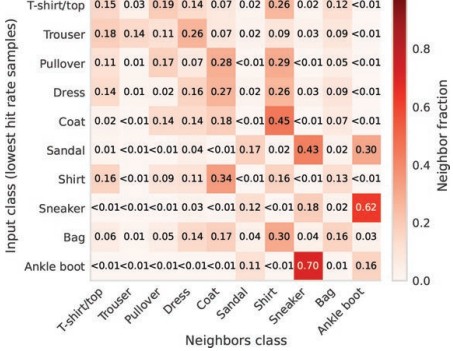

Figure 11: Neighbor confusion matrix (FashionMNIST) inputs with the lowest neighbor label hit rate (worst 5k samples). Each row corresponds to an input class and each column to a neighbor class; entry $(i, j)$ gives, averaged over low-hit inputs of class $i$, the fraction of their $k$-NN that belong to class $j$.

deformation). In effect, our method thrives in latent spaces with clear separation between classes, and is not required to have small within-class variations in the data.

Input $x$           Nearest neighbors $k$-NN($x$) ($k = 10$)

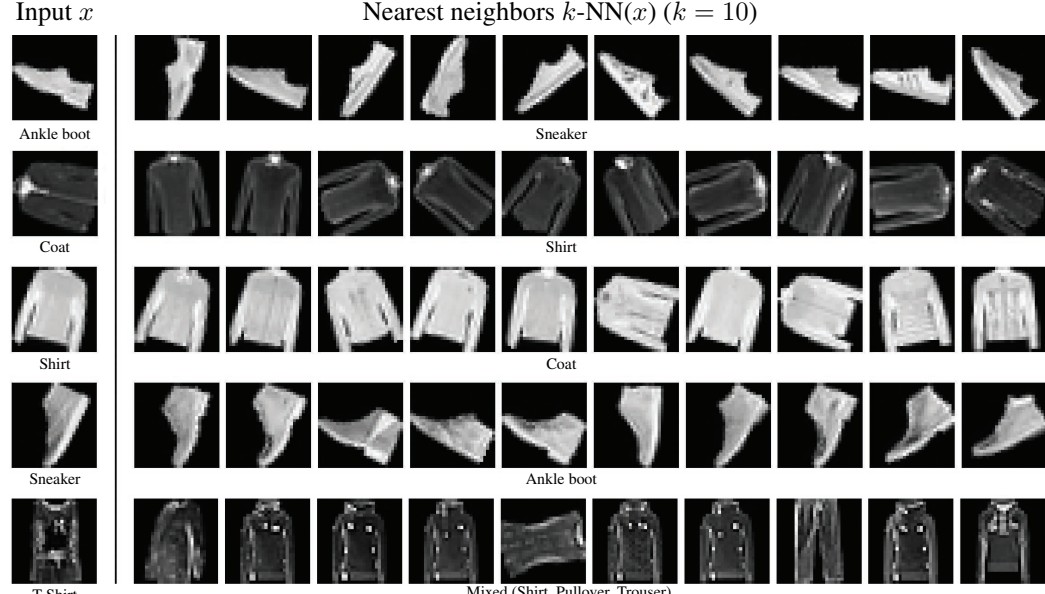

Figure 12: Qualitative analysis of lowest hit rate equivalence classes in FashionMNIST. Each row shows a training sample (left) with low hit rate (in particular, zero for rows 1-4) and its 10 nearest neighbors in the learned latent space (right), ordered by cosine similarity. Text underneath denotes labels. This illustrates inter-class feature overlap in the latent space: some inputs from different semantic classes exhibit very similar features (e.g., some *Sneaker* models look very similar to some *Ankle boot* models), therefore they are clustered together despite having different labels. This overlap is the primary cause of reduced hit rates in FashionMNIST for certain inputs, and highlights the regime where our equivalence class definition is most challenged.

As an empirical example, GEOM-QM9 exhibits non-trivial intra-class conformational variations (RMSD $\leq 1.5$ Angstrom) yet achieves a $0.95$ hit rate (cf. Figure 13a). This happens because molecular identity has a very strong and clear signal from compositional and topological invariants, e.g. number of atoms of a given element, that remain perfectly constant across conformers, which allows the encoder to more easily separate conformers of different molecules. Our equivalence class definition behaves well as a result.

**Mitigation strategies for inter-class overlap**    Mitigation strategies for these cases should therefore target inter-class overlap, i.e., increasing class separation in the invariant latent space. This challenge is well-studied, and established approaches with contrastive or self-supervised objectives (e.g., DINO (Caron et al., 2021), BYOL (Grill et al., 2020)) are known to sharpen class boundaries without supervision. Such objectives are perfectly compatible with our framework and directly address its main failure mode. Additional mitigation strategies, as discussed in our limitations section, include richer and more robust neighborhood sampling to further stabilize equivalence class construction.

### D.4 PICKING UP ON PARTIAL SYMMETRIES

A notable strength of RECON is its ability to distinguish between *distinct* classes that are related by a group transformation – such as digits '6' and '9', which are related by a $180°$ rotation. Fully $SO(2)$-equivariant methods map these inputs to an equivalent representation, and therefore, downstream tasks struggle distinguishing between them. This is a well-known example that has motivated partially equivariant methods in the past (Romero & Lohit, 2022). RECON leverages the clustering of the input's *invariant features* and normalizes their pose distributions separately, which addresses this problem. Other class-pose decomposition methods, much like classical equivariant networks, can not pick up on partial symmetries, and therefore collapse both '6's and '9's into the same canonical

Table 1: RECON pseudo-label generation (Algorithm 1) wall-clock runtime for different datasets (naive implementation).

| Dataset | Number of samples | Nearest neighbors | Runtime (seconds) |
|---|---|---|---|
| MNIST | 12,000 | 10 | 15 |
| Fashion-MNIST | 60,000 | 10 | 173 |
| GEOM-QM9 | 174,481 | 25 | 1033 |

reconstruction (Figure 5b Col. 2).[7] Consequently, the distributions $\nu_{[6]}$ and $\nu_{[9]}$ of relative poses are different with opposing peaks (Figure 9 Col. 2, Digits 6 and 9). On the contrary, RECON estimates distinct offsets ($\hat{\Gamma}_{[6]}$ and $\hat{\Gamma}_{[9]}$) based on how each digit class typically appears relative to the arbitrary canonical. As a result, normalization yields not only a *shared* symmetry pattern ($\mu_{[6]} \approx \mu_{[9]}$, Figure 9 Col. 3) but also correctly associates them with *distinct* natural poses (Figure 5b Col. 3). This sensitivity to the data's contextual orientation allows RECON to handle cases beyond perfect group orbits where full equivariance is inappropriate.

### D.5 Computational analysis

Our method introduces three computational components on top of the class-pose backbone: (i) a (one time) pseudo-label generation step based on $k$-nearest neighbors (Algorithm 1), (ii) training of $\Theta$ and $\Lambda$ (computation and implementation details in Appendix E) and (iii) the downstream symmetry/OOD/canonicalization inferences with the learned mappings $\Theta$ and $\Lambda$. We analyze (i) and (iii) in terms of complexity and runtime in practice.

**Pseudo-label generation** Let $N$ be the number of training examples, $d$ the dimensionality of the invariant embedding $z$, and $k$ the number of neighbors for Algorithm 1. The dominant cost of pseudo-label generation is the $k$-NN computation, with complexity $\mathcal{O}(N^2 d)$. This step is ran once per dataset, and the resulting pseudo-labeled dataset is saved to disk and subsequently used to train $\Theta$ and $\Lambda$. We emphasize that the pseudo-label generation is a one-time computation that does not happen at training or inference time. On our hardware (see Appendix E), a naive, non-optimized implementation of the pseudo-label generation process completes in the order of few seconds / minutes per dataset; wall-clock times are reported in Table 1. For scaling to substantially larger $N$, once can use batch processing to reduce memory and replace the $k$-NN search by highly optimized neighbor search implementations (e.g., FAISS (Douze et al., 2024)), which greatly reduces the effective cost.

**RECON canonicalization overhead** Canonicalization at test time makes use of two forward passes: the IE-AE encoder on the input to obtain the relative pose $\psi(x)$ (which yields the IE-AE canonicalization), and the forward pass $\Lambda(x)$ to obtain the centering transformation (which alongside $\psi(x)$ yields the RECON canonicalization, $C(x) = \rho_\mathcal{X}(\Lambda(x) \cdot \psi(x)^{-1}) x$). We quantify RECON's canonicalization overhead by measuring the average wall-clock inference time (in ms/sample) of the centering transformation computation. The computational overhead is of 0.351 ms/sample for MNIST, 0.113 ms/sample for FashionMNIST and 0.068 ms/sample for GEOM-QM9. Our canonicalization adds only a small overhead, and the performance gains outweigh this cost.

## E Implementation details of RECON and IE-AE backbone

Our experiments are implemented using Python 3.11, primarily with Pytorch (Paszke et al., 2019) and Pytorch Geometric (Fey & Lenssen, 2019) for neural network implementation and training. All experiments were seeded with a fixed random seed for reproducibility and logged using Weights & Biases (Biewald, 2020).

---

[7]This limitation can also be observed in SGM (Allingham et al., 2024).

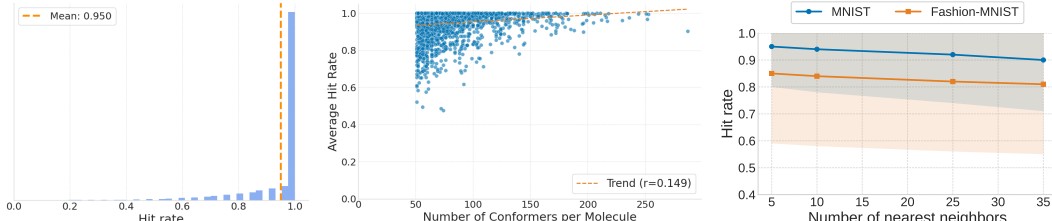

(a) Per-input hit rate in GEOM-QM9, measuring quality of the computed equivalence classes $[x]$.

(b) Average hit rate per-molecule vs number of conformers. Molecules with less conformers present noisier equivalence classes.

(c) Average and standard deviation of hit rate across neighbors in imaging experiments.

Figure 13: Quantitative analysis of the proposed equivalence class definition for the GEOM-QM9 experiment.

### E.1 IMAGES

#### E.1.1 DATA PREPARATION

We use augmented versions of the MNIST/FashionMNIST datasets with different true symmetry distributions for each class. We load the original training and testing sets, split images by their class label (0-9), and apply a rotation to each image sampled from a label-dependent distribution (e.g. uniformly from ($[-60°, 60°]$ for labels 0-4, and $[-90°, 90°]$ for labels 5-9 in MNIST).

#### E.1.2 MODEL IMPLEMENTATIONS

For the equivariant architectures, we use the `escnn` library (Weiler & Cesa, 2019; Cesa et al., 2022).

**Encoder.** The encoder processes the input image through a series of $SE(2)$-equivariant blocks (`escnn.nn.R2Conv` layers with kernel sizes 7, 5, 5, 5, 5, 3, 1, using `gspaces.rot2dOnR2`), with additional batch normalization and non-linearities (`ReLU`, `NormNonLinearity`).. The final layer outputs features decomposed into invariant scalar fields (128 channels, representing the invariant component $\eta(x)$), and equivariant vector fields from which the relative pose $\psi(x)$ – parameterized by an angle – is derived.

**Decoder.** A standard CNN takes the 128-dimensional invariant latent vector as input. It uses a sequence of `torch.nn.Conv2d`, `BatchNorm2d`, `Dropout2d` ($p$=0.2), and `ReLU` layers, along with bilinear interpolation for upsampling, to reconstruct the input image. The output passes through a Sigmoid activation.

**Learnable mappings $\Theta$ and $\Lambda$.** First, $\Theta$ computes an invariant embedding from the input using `escnn.nn.R2Conv` layers as before. Then, this embedding is passed through a small MLP consisting of two `torch.nn.Linear` layers with ReLU activations, predicting a single scalar, parameter of the target distribution (an angle $\hat{\theta}$ for MNIST experiment and a standard deviation angle $\hat{\sigma}$ for FashionMNIST experiment). $\Lambda$ is built equivalently as the $\Theta$ predictor, but outputs a single scalar value representing the predicted transformation offset $\hat{\Gamma}$ parameterized as an angle.

#### E.1.3 TRAINING

- **Training the IE-AE:** Best hyperparameter configuration (as in configuration that yields the lowest reconstruction error during validation) was found by hyperparameter tuning. The IE-AE components were trained for 700 epochs using the Adam optimizer (Kingma & Ba, 2017) with a learning rate $\approx 8 \times 10^{-4}$ and a batch size of 128. We save model weights corresponding to the lowest validation reconstruction loss.

- **Canonical orientation normalization:** Using the frozen pre-trained encoder, we compute the invariant embedding and the relative rotation angle degrees for each training sample. We then compute the $k = 10$ nearest neighbors (for each sample) in the $\eta$ space based on cosine

similarity . Finally, for each sample $x$, we compute two pseudo-labels for training $\Lambda$ and $\Theta$ respectively:

- $\Gamma_{[x]}$: the Fréchet mean (an angle in degrees) – which is equivalent to the circular mean in $SO(2)$ – of the set of neighbor angles $\{\psi(x)\}_{x \in \mathcal{NN}(x)}$.

- $\hat{\theta}_{[x]}$: the set of neighbor angles $\{\psi(x)\}_{x \in \mathcal{NN}(x)}$ is then normalized using the previous Fréchet mean: $\{\psi(x)\Gamma_{[x]}^{-1}\}_{x \in \mathcal{NN}(x)}$. From this set of normalized angles, the pseudo-label $\hat{\theta}_{[x]}$ (estimate of the parameter of the symmetry distribution) is calculated using standard parameter estimation methods. In our case, we use methods based on moments robust to outliers.

- **Training the learnable mappings $\Theta$ and $\Lambda$:** The $\Theta$ predictor and $\Lambda$ predictor were trained jointly for 600 epochs by minimizing MSE between the network outputs and the pseudo-labels. We used a combined Adam optimizer targeting the parameters of both predictors, with a learning rate of approximately $1.35 \times 10^{-4}$. We use a batch size of 128. We weight both losses by a weighting factor of 0.25 applied to the loss of $\Lambda$. We save model weights corresponding to the best validation loss.

For all our trainings, we employ a cosine annealing learning rate scheduler with a warm-up phase of 5 epochs and restarts. Trainings were performed using an NVIDIA A100-SXM4-80GB graphics cards, running for approximately $6 + 1$ hours for MNIST and $20 + 3$ hours for FashionMNIST (IE-AE + learnable mappings phase).

## E.2 MOLECULAR CONFORMATIONS

### E.2.1 DATA PREPARATION AND SELECTION

Our starting point is the GEOM dataset (Axelrod & Gómez-Bombarelli, 2022), focusing on its QM9 subset (Ramakrishnan et al., 2014). From this, we select molecules that possess at least 64 distinct conformers. We focus on low-energy states and only retain conformers with a Root Mean Square Deviation (RMSD) of less than 1.5 Angstrom from their respective molecule's minimum-energy conformer. This threshold is applied using RDKit's (Landrum et al., 2025) `rdkit.Chem.rdMolAlign.GetBestRMS` function, which also aligns each qualifying conformer to its corresponding minimum-energy reference structure. This alignment step provides an orientation-neutral base for each set of conformers.

The conformers for each selected molecule are then randomly split into training, validation, and test sets with an $0.8, 0.1, 0.1$ ratio (*per molecule*) respectively. This results in $174, 481$ training samples, $20, 902$ validation samples and $23, 805$ test samples across $2, 221$ distinct classes of molecules.

To introduce controlled and diverse controlled global orientations to serve as ground truths, we augment the data by applying random rotations to the aligned conformers. These rotations are sampled from matrix-Fisher distributions (Mardia & Jupp, 2009), a unimodal distribution on $SO(3)$ suitable for modeling varied directional concentrations.

Specifically, we utilize three distinct matrix-Fisher parameter matrices $F_{true}$ that simulate rotations around the standard $e_1, e_2, e_3$ axes in the 3D space:

- $F_{true}^1 = \text{diag}(100, 0.001, 0.001)$
- $F_{true}^2 = \text{diag}(0.001, 100, 0.001)$
- $F_{true}^3 = \text{diag}(0.001, 0.001, 100)$

Each molecule, along with its conformers, is randomly assigned one of these $F_{true}$ matrices. Rotations are then sampled for each conformer from its assigned $F_{true}$ matrix. This process allows us to simulate distinct, realistic and parametrically defined orientation preferences across different molecules in the dataset.

### E.2.2 DATA PRE-PROCESSING

In our graph-based framework, molecules are represented as graphs $\mathcal{G} = (\mathcal{V}, \mathcal{E})$, where nodes $\mathcal{V}$ correspond to atoms and edges $\mathcal{E}$ represent connections between them. The Steerable E(3)-Equivariant

Graph Neural Network (SEGNN) (Brandstetter et al., 2022) processes these graphs, leveraging their geometric and chemical information.

**Initial node features.** Each atom $i \in \mathcal{V}$ is initially characterized by a feature vector $\mathbf{x}_i$. This vector is a one-hot encoding of the atom type (e.g., Hydrogen, Carbon, Nitrogen, Oxygen, Fluorine). For our SEGNN model, this corresponds to an input irreducible representation (irrep) of $5 \times 0e$, representing five distinct scalar atom types.

**Graph connectivity and edge definition.** The graph's edges are determined using a radius graph approach. An edge $(i, j)$ exists between atom $i$ and atom $j$ if their Euclidean distance is within a predefined cutoff radius $r_{cut}$. In our experiments, we use $r_{cut} = 2.0$. The connectivity is stored in `edge_index` as per PyTorch Geometric conventions.

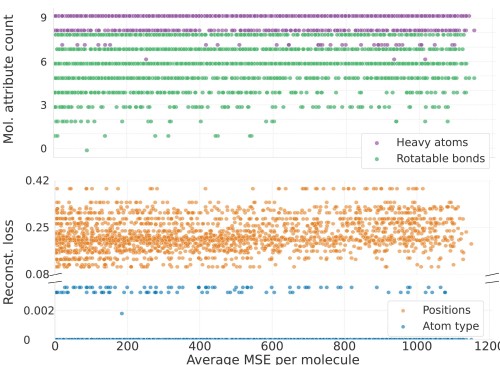

Figure 14: MSE vs reconstruction error and molecule attributes.

**Edge attributes.** To incorporate 3D geometry in an equivariant manner, edges are augmented with attributes derived from the relative positions of the connected atoms (Brandstetter et al., 2022). Specifically, for an edge $(i, j)$ connecting atom $i$ at position $\mathbf{p}_i \in \mathbb{R}^3$ to atom $j$ at position $\mathbf{p}_j \in \mathbb{R}^3$, we compute the relative position vector $\mathbf{r}_{ij} = \mathbf{p}_i - \mathbf{p}_j$. These relative position vectors are then transformed into spherical harmonics up to a maximum degree $l_{max}^{edge}$ (we use $l_{max}^{edge} = 3$). We use the `e3nn` (Weiler et al., 2018; Thomas et al., 2018; Kondor et al., 2018) library to compute spherical harmonics $\mathbf{Y}(\mathbf{r}_{ij})$, which serve as edge attributes for the SEGNN. The raw relative squared Euclidean distance $d_{ij}^2 = ||\mathbf{r}_{ij}||^2$ is also added and passed to the SEGNN as an additional scalar $1 \times 0e$ type feature for each edge.

**Node attributes.** In addition to the initial atom type features, nodes are also assigned geometric attributes based on the mean of the spherical harmonic attributes of their incoming edges. These node attributes are processed by the SEGNN.

### E.2.3 MODEL IMPLEMENTATION

**SEGNN-based encoder.** The SEGNN architecture learns equivariant node representations through message passing. Our implementation consists of 6 message passing blocks. Each layer operates on node features represented by a combination of irreducible representations (irreps): we use 100 scalar channels (type 0e irreps), 64 vector channels (type 1o irreps), 16 type 2e tensor channels, and 8 type 3o tensor channels. Equivariant tensor products and gated Sigmoid Linear Units (SiLU) non-linearities are used throughout these layers to ensure equivariance. Instance normalization is applied to the features within the SEGNN layers.

After the main message passing sequence, a final equivariant projection layer maps the processed node features to a target set of $d_{inv} = 512$ scalar channels and $d_{eq} = 512$ vector (type 1o) channels per node. These structured node-level features are then processed as:

1. **Pooling:** The scalar ($0e$) and vector ($1o$) components of the node features are independently pooled across all nodes to obtain graph-level invariant and equivariant features (per-graph: an invariant embedding $\mathbf{s}_{pool} \in \mathbb{R}^{512}$ and an equivariant feature set $\mathbf{v}_{pool}$ consisting of 512 3D vectors). Additionally, the invariant embedding is passed through a Multi-Layer Perceptron (linear layers with ReLU activations) to produce a final graph-level invariant latent encoding $\mathbf{z}_{inv} \in \mathbb{R}^{512}$.

2. **SE(3) pose prediction:** The equivariant embedding must be processed to obtain group transformations $(R, \mathbf{t}) \in SE(3)$. We pass the pooled equivariant features $\mathbf{v}_{pool}$ through an equivariant MLP to obtain three equivariant 3D output vectors: $\mathbf{y}_1, \mathbf{y}_2$ and $\mathbf{y}_t$. A rotation

matrix $R \in SO(3)$ is then derived from $\mathbf{y}_1$ and $\mathbf{y}_2$ using a Gram-Schmidt orthogonalization procedure. The translation vector $\mathbf{t} \in \mathbb{R}^3$ is simply given by $\mathbf{y}_t$.

This construction leads to an embedding $\mathbf{z}_{inv}$ invariant to $SE(3)$ transformations of the input graph, and to a predicted pose $(R, \mathbf{t}) \in SE(3)$ that transforms in an equivariant manner.

**Decoder.**   The decoder reconstructs the molecule from the invariant latent code $\mathbf{z}_{inv}$. It comprises two main components: a position decoder and an atom-type decoder. Both are implemented as MLPs with residual blocks and ReLU activations.

1. The **Position decoder** takes $\mathbf{z}_{inv}$ as input and outputs a set of 3D coordinates for a maximum number of 29 atoms in our dataset. This set of 3D coordinates represents the molecule in a learned canonical orientation.

2. The **Atom-type decoder** also takes $\mathbf{z}_{inv}$ and predicts the logits for atom types for each of the 29 positions.

The final reconstructed atom positions are obtained by applying the predicted equivariant transformation $(R, \mathbf{t})$ to the canonical positions obtained by the decoder.

**Learnable mappings $\Theta$ and $\Lambda$.**   During this self-supervised learning phase we train two separate networks using the computed pseudo-labels as outlined in Section 3.3. Both networks use the same architectural pattern as the encoder: an SEGNN backbone to generate an invariant graph-level embedding, followed by an MLP head.

The $\Theta$ network outputs 9 parameters to form the $3 \times 3$ matrix-Fisher parameter matrix $F_{pred}$. The $\Lambda$ network outputs a rotation matrix representing the predicted offset $\hat{\Gamma}$.

### E.2.4   TRAINING

- **Training the IE-AE:** The encoder and a decoder based on SE(3)-equivariant graph neural networks as defined previously were trained for 600 epochs using the Adam optimizer with a learning rate of $\approx 8.89 \times 10^{-5}$ and a batch size of 128. We compute two loss functions for each of the decoder outputs – molecule's positions and molecule's atom types respectively.

  - A positional reconstruction loss, computed as an L1 loss (mean absolute error) between the true node coordinates and the coordinates obtained by applying the predicted transformation $(R, \mathbf{t})$ to the decoder's output positions.

  - An atom-type reconstruction loss, which is a cross-entropy loss between the true atom types and the atom types predicted by the decoder.

  The loss contribution by the atom-type loss is weighted by a factor of $\approx 3.614$. We save model weights (both encoder and decoder) corresponding to the lowest validation positional reconstruction loss.

- **Canonical orientation normalization:** Using the frozen pre-trained encoder, we compute the invariant embedding and the observed relative rotation matrix for each training sample (molecule). We then compute the $k = 25$ nearest neighbors for each sample in the $\eta$ space based on cosine similarity. Finally, for each sample $x$ we compute two pseudo-labels for training the $\Lambda$ and $\Theta$ predictors respectively:

  - $\hat{\theta}$: The set of neighbor rotation matrices $\{\psi(x)\}_{x \in \mathcal{NN}(x)}$ is then normalized using the inverse of the centering rotation as $\{\psi(x)\hat{\Gamma}_{[x]}\}_{x \in \mathcal{NN}(x)}$. From this set of normalized rotation matrices, we estimate the parameters of the matrix-Fisher distribution $(\hat{\theta} = \hat{F})$ on these aligned rotations via the moment-matching approach (inverting $A(s) = \coth(s) - 1/s$ by Newton's method and reconstructing $\hat{F}$ via SVD) (Wood, 2008; Mardia & Jupp, 2009). This $\hat{\theta}$ is the pseudo-label for the $\Theta$ network.

  - $\hat{\Gamma}_{[x]}$: This is the Fréchet mean on $SO(3)$, estimated as the mode of the matrix-Fisher distribution fitted to the set of observed neighbor rotations $\{\psi(x)\}_{x \in \mathcal{NN}(x)}$. We compute it using the SVD-based moment matching estimator for the mode standard

in literature (Wood, 2008; Mardia & Jupp, 2009). This mode $\hat{\Gamma}_{[x]}$ then serves as the pseudo-label for the $\Lambda$ network.

- **Training the learnable mappings $\Theta$ and $\Lambda$:** The $\Theta$ and $\Lambda$ predictor were trained jointly for 150 epochs. The SEGNNs for these predictors use 4 layers, with $50\times0e + 32\times1o + 8\times2e + 4\times3o$ hidden irreps, and instance normalization. We minimized the MSE between the network outputs and their respective pseudo-labels. A combined Adam optimizer was used for the parameters of both predictors, with a learning rate of approximately $4.83 \times 10^{-4}$ and a batch size of 128. The loss contribution from the $\Lambda$ predictor was weighted by a factor of 500. We save model weights to best validation loss.

For all our trainings, we employ a cosine annealing learning rate scheduler with a warm-up phase of 5 epochs. Gradient clipping with a maximum norm of 1.0 is applied during the training of the learnable mappings phase. Training was performed using an NVIDIA A100-SXM4-80GB graphics cards, running for approximately $6 + 1.5$ days (IE-AE + learnable mappings phase).

## F    IMPLEMENTATION DETAILS OF DOWNSTREAM APPLICATIONS

### F.1    OOD DETECTION

The distributions recovered by RECON can be used to identify objects in unnatural (out-of-distribution) poses relative to their learned symmetry profile. Given predictors $\Theta(x)$ and $\Lambda(x)$ (Sec. 3.3), consider the absolute pose as $g_{abs} = \psi(x)\Lambda(x)^{-1}$ (normalized relative pose). The likelihood of $g_{abs}$ under the distribution parameterized by $\Theta(x)$ serves as an anomaly score,

$$s(x) := -\log p_{\Theta(x)}(g_{abs}). \tag{23}$$

Low $s(x)$ indicates in-distribution, while high $s(x)$ indicates OOD. We empirically validate this by classifying randomly oriented $SO(2)$ (images) and $SO(3)$ (GEOM) test instances.

Theoretically, one could skip RECON centering and score with an uncentered class-pose density using arbitrary canonicals, i.e., using the score $s_{\mathrm{rel}}(x) := -\log p_{\tilde{\Theta}(x)}(\psi(x))$ where $\tilde{\Theta}(x)$ is trained on pseudo-labels derived from raw relatives $\{\psi(x)\}$. In effect, we show (Proposition B.2) that the log density (and therefore the proposed score $s(x)$) is invariant to the choice of reference frame. However, in practice, centered distributions (RECON) perform better. Centered distributions facilitate optimization: they concentrate probability mass near the identity (reducing variance and dynamic range of the loss), decouple the center $\hat{\Gamma}_x$ from the shape parameters (which are learned by $\Theta$), and overall result in more stable training. On the contrary, uncentered distributions introduce class-specific, arbitrary shifts $\hat{\Gamma}_{[x]}$ that must be jointly learned by $\Theta$, increasing complexity. Empirically this results in consistently higher AUC-ROC across datasets for RECON (Table 5; cf. IE-AE vs. RECON, ROC curve plots in Fig. 15).

We now offer a detailed implementation and discussion of the OOD experiment, including dataset preparation, metrics and uncentered (IE-AE) ablation.

#### F.1.1    IMAGING

**Pre-processing and metrics.** Each MNIST test image is rotated by a random angle $\theta \sim \mathcal{U}(-180°, 180°)$. An input is labeled in-distribution (ID) if its applied rotation lies within the class-specific training symmetry range (digits 0-4: $\pm60°$; digits 5-9: $\pm90°$), and out-of-distribution (OOD) otherwise. All angular differences are computed on $(-180°, 180°]$; we write $\mathrm{angdiff}(\alpha, \beta) = \mathrm{wrap}(\alpha - \beta)$.

We aim to verify that our anomaly score $s(x)$ assigns larger values to OOD inputs than to ID. For this, we report AUC-ROC, Area Under the Receiver Operating Characteristic Curve (using `sklearn`'s `sklearn.metrics.roc_auc_score`), which summarizes performance *across all decision thresholds* $\tau$ (if we classify $x$ as OOD when $s(x) \geq \tau$). We use AUC-ROC since it is a threshold-free measure and works well in class imbalance scenarios, a property that other metrics (e.g., accuracy) lacks (they require a fixed threshold, whose choice depends on the use case for deployment, e.g. if one wants to optimize threshold for minimizing false positives). We also provide AUC-ROC curve plots in Fig.15.

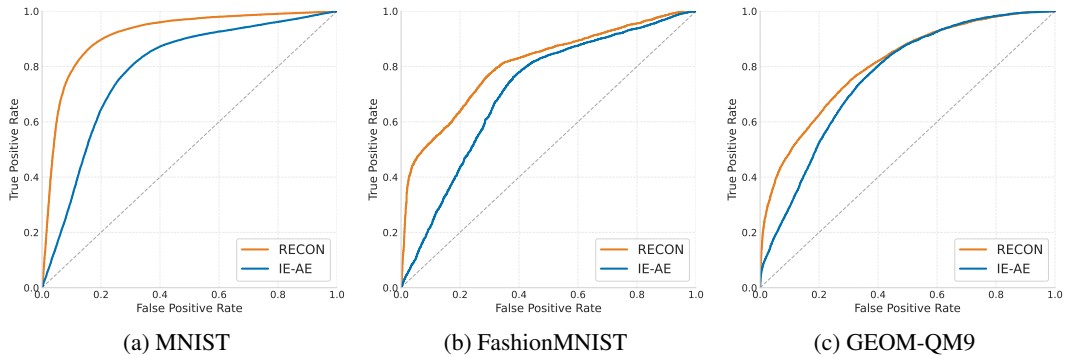

(a) MNIST        (b) FashionMNIST        (c) GEOM-QM9

Figure 15: ROC curves for IE-AE and RECON-based anomaly scores for (a) MNIST, (b) FashionM-NIST, and (c) GEOM-QM9 experiment.

**Uncentered distributions (baseline, IE-AE)**   Given $x$, the model predicts an uncentered uniform support $\tilde{\Theta}(x) = (a_x, b_x)$ (in degrees). Let the input's relative pose be $g_{\mathrm{rel}} = \mathrm{wrap}(\psi(x))$ and the support midpoint $\mu_x = \mathrm{wrap}\left(\frac{a_x + b_x}{2}\right)$. We use the absolute deviation

$$s_{\mathrm{rel}}(x) \;=\; \big|\mathrm{angdiff}(g_{\mathrm{rel}}, \mu_x)\big|. \tag{24}$$

(Equivalently, one may use distance to the predicted support,

$$s_{\mathrm{rel}}(x) = \max\Big\{0, \big|\mathrm{angdiff}(g_{\mathrm{rel}}, \mu_x)\big| - w_x\Big\}, \qquad w_x = \tfrac{1}{2}\,\mathrm{wrap}(b_x - a_x). \tag{25}$$

Both are surrogates of the uniform negative log likelihood, differing only by a within-support constant.)

**Centered distributions (RECON)**   Let $\Gamma_x$ be the predicted centering transformation (in degrees) from $\Lambda(x)$, and define the absolute (centered) pose $g_{\mathrm{abs}} = \mathrm{wrap}\big(g_{\mathrm{rel}} - \Gamma_x\big)$. For identity-centered supports, the anomaly score reduces to

$$s(x) \;=\; |g_{\mathrm{abs}}|. \tag{26}$$

For FashionMNIST, we mirror the MNIST setup, but assign ID/OOD labels using the FashionMNIST class-specific symmetry ranges specified in Section 5. No other changes are made.

### F.1.2   GEOMETRIC GRAPHS

**Pre-processing and metrics.**   We start from the (augmented) GEOM-QM9 test set as described in Section 5 and group conformers by molecule (SMILES). For each molecule, we randomly split its conformers into two halves: one half is kept as in-distribution (ID) and left unchanged; for the other half we generate OOD *candidates*. This split avoids a trivial task: because $SO(3)$ is large, naively applying a random rotation to every conformer would produce an overwhelming fraction of OOD examples.

For OOD candidates, we first *canonicalize* by undoing the augmentation used to create the conformer (apply the rotation used to create the augmented sample $R_{\mathrm{aug}}$, to bring coordinates to the original canonical frame), and then apply a fresh random rotation $R_{\mathrm{candidate}} \sim \mathrm{Haar}(SO(3))$ to obtain the new coordinates. ID examples are those left unmodified.

Ground-truth labels for the candidate half are decided through likelihood: if the sample rotation $R_{\mathrm{candidate}}$ has a high likelihood of coming from $F$ (the molecule's matrix-Fisher parameter that was assigned during the creation of the augmented dataset), then that sample is labeled as ID. Otherwise, the sample is labeled as OOD.

Note that since the matrix-Fisher density on $SO(3)$ is proportional to $\exp(\mathrm{tr}(F^\top R))$, then the unnormalized log-likelihood of $R_{\mathrm{candidate}}$ is $\mathrm{tr}\big(F^\top R_{\mathrm{candidate}}\big)$ up to an additive constant. Therefore, we mark a candidate as OOD if $\mathrm{tr}\big(F^\top R_{\mathrm{candidate}}\big) < \tau$ (low likelihood), where $\tau$ is a fixed threshold controlling class balance (we use $\tau = 96$).

**Uncentered distributions (baseline, IE-AE)**   Let $\psi(x) \in \mathrm{SO}(3)$ be the predicted pose and let $F(x) \in \mathbb{R}^{3\times3}$ be the matrix-Fisher parameter predicted from $x$ using the learned mapping. We score anomalies with the negative unnormalized log likelihood

$$s_{\mathrm{rel}}(x) \;=\; -\langle F(x),\, \psi(x)\rangle \;=\; -\operatorname{tr}\!\big(F(x)^\top \psi(x)\big),$$

where $\langle A, B\rangle := \operatorname{tr}(A^\top B)$. When $F(x)$ is axis-symmetric of the form $F(x) = \kappa(x)\,\mu_x$ with mode $\mu_x \in \mathrm{SO}(3)$, $s_{\mathrm{rel}}(x)$ is a monotone surrogate of the geodesic deviation $\phi\big(\mu_x^{-1}\psi(x)\big)$.

**Centered distributions (RECON)**   Let $\Lambda(x) \in \mathrm{SO}(3)$ be the predicted centering transform and define the absolute (centered) pose

$$g_{\mathrm{abs}} \;=\; \psi(x)\,\Lambda(x)^{-1}.$$

We then score

$$s(x) \;=\; -\langle F(x),\, g_{\mathrm{abs}}\rangle \;=\; -\operatorname{tr}\!\big(F(x)^\top g_{\mathrm{abs}}\big),$$

a monotone surrogate of the geodesic angle $\phi\big(g_{\mathrm{abs}}\big)$.

### F.2   GRANTING GROUP INVARIANCE TO PRE-TRAINED BACKBONES

#### F.2.1   IMAGING

We evaluate inference-only canonicalization using a classifier (ResNet18 backbone) trained on the raw (no augmentations) MNIST and FashionMNIST datasets. We then evaluate the performance of this pre-trained model on the rotated dataset variations created for the symmetry discovery experiment outlined in Section 5 (that is, $\pm60°$ / $\pm90°$ rotations for MNIST, etc). For both MNIST and FashionMNIST, we train a ResNet-18 with cross-entropy and Adam optimizer (lr=$1e-3$, batch size 128, 100 epochs) and save the best classifier checkpoint based on best validation accuracy. At test time, we compare three input pre-processing modes before feeding images into the classifier: (i) no canonicalization (using the pre-trained classifier), (ii) arbitrary canonicalization (taken from the IE-AE) and (iii) RECON canonicalization. As a reference value, we additionally report accuracy on the original (un-augmented) test set (Table 2), which matches the training distribution by construction. This corresponds to a perfect canonicalization (reversing the augmentations), and serves as an upper bound of the best accuracy that can be obtained if we provide a perfect canonicalization function during inference.

**Comparisons**   We compare with EquiAdapt (Mondal et al., 2023), another test-time canonicalization method. We reproduce their *zero-shot setup*, that is, only the canonicalization function is trained, and the pre-trained model weights are kept frozen (same setting as RECON). We use the $SO(2)$ canonicalization prior and steerable network from their public `equiadapt` package for the canonicalization function. The canonicalization function is attached on top of the pre-trained ResNet18 classifier, and is trained for 50 epochs with a batch size of 128. We choose the best hyperparameter configuration based on best test accuracy (see hyperparameter search in Figure 4).

For reference, Table 3 reports results for specialized group equivariant (and partially equivariant) architectures trained from scratch on the same non-rotated datasets. Equivariant models are by construction designed to be robust to symmetry-induced test-time distribution shifts. On MNIST, our canonicalization (RECON, $90.96\%$) shows a moderately small gap w.r.t. the $SE(2)$-equivariant ESCNN classifier ($94.72\%$), while keeping the benefits of operating purely as an unsupervised, test-time, plug-and-play module for arbitrary classifiers. On FashionMNIST however, RECON actually performs better than the fully equivariant ESCNN. Lastly, we consistently outperform Partial G-CNNs in both datasets, sometimes by a large margin ($67.72\%$ vs $81.96\%$ in FashionMNIST).

The ESCNN classifier is built using the same architecture as the ESCNN backbone for the IE-AE described in Appendix E.1, but with a 2-layer MLP classification head attached at the end. The invariant embedding dimension before the classification head is of size $1024$ and the ESCNN backbone has a hidden dimension of $128$, totaling $3,384,463$ trainable parameters in this case. We train for 50 epochs with batch size 128 and learning rate $1e-3$. Hyperparameter configuration was chosen based on best test accuracy across different learning rates, hidden dimensions and embedding dimensions (Table 5). The Partial G-CNNs were configured and trained similarly, aiming to keep an approximate same number of parameters as the ESCNN architectures.

Table 2: Test accuracy on the in-distribution test set (i.e., non-augmented) obtained with the pre-trained classifier trained on non-rotated, vanilla datasets: reference canonicalization upper bound representing a perfect canonicalization function.

| Dataset | Augmentation-reversed test set acc. (upper bound, perfect canon.) |
|---|---|
| MNIST | $98.14 \pm 0.02\%$ |
| FashionMNIST | $91.11 \pm 0.1\%$ |
| GEOM-QM9 | $96.40 \pm 0.0\%$ |

Table 3: Test accuracy on the per-class rotated datasets obtained with fully and partially equivariant classifiers trained on non-rotated, vanilla datasets; auxiliary reference metric representing the performance obtained with specialized equivariant architectures that are, by construction, robust to symmetry-induced distribution shifts during inference.

| Dataset (Train / Test) | Group | Equivariance | Backbone | Test set acc. |
|---|---|---|---|---|
| MNIST (Orig / Rot) | $SE(2)$ | Full | ESCNN | 94.59% |
|  |  | Learned | Partial G-CNN | 90.20% |
| FashionMNIST (Orig / Rot) | $SE(2)$ | Full | ESCNN | 77.94% |
|  |  | Learned | Partial G-CNN | 67.72% |
| GEOM-QM9 (Orig / Rot) | $SE(3)$ | Full | SEGNN | 98.55% |

### F.2.2 GEOMETRIC GRAPHS

We evaluate inference-only canonicalization on GEOM-QM9 molecular graphs. We first train a graph CNN on aligned conformers only: a 3 layer GCN (using Pytorch Geometric's `GCNConv` layer with a 128 hidden size) followed by a a two layer MLP head (64 hidden units, dropout 0.1), trained with cross-entropy and Adam (batch size 128 for 50 epochs), saving the best checkpoint by validation accuracy. Node inputs to the classifier are geometric attributes based on the mean of the spherical harmonic attributes of their incoming edges (see details in data pre-processing section of GEOM-QM9 dataset in Appendix E.2.2). At test time, we evaluate on the rotated conformer test set described in Appendix E.2.1, and compare three input pre-processing modes applied per graph before classification: (i) none (no canonicalization, using the pre-trained classifier), (ii) arbitrary canonicalization (taken from a the IE-AE) and (iii) RECON canonicalization. Note that after rotation by either canonicalization method, we have to recompute the geometric features (edge/node attributes) from the updated coordinates, since those are the input to the pre-trained GCN. For a reference on the upper bound of a perfect canonicalization, we also report accuracy on the test dataset obtained after reversing the test-set augmentations (Table 2). All runs use a fixed global seed with deterministic settings.

**Comparisons** For comparison against EquiAdapt (Mondal et al., 2023), we attach and train their continuous $SO(3)$ point-cloud canonicalizer from the public `equiadapt` package on top of the pre-trained GCN. For each input molecular conformation, we canonicalize its atomic coordinates with the canonicalizer, and then compute the resulting conformation's geometric features (as in Appx. E.2.2) for input into the frozen GCN (same approach as in our test-time canonicalization). We use Adam with learning rate $1e-3$, batch size 128, and train for 50 epochs. During training, the GCN weights are frozen and only the canonicalizer is optimized (coined the *zero-shot setup* in EquiAdapt (Mondal et al., 2023), equivalent to our test-time canonicalization setup), as opposed to the alternative *fine-tuning setup* in which the pre-trained model is retrained.

On GEOM-QM9, the $SO(3)$-equivariant SEGNN reaches 98.55% accuracy (Table 3), far above the pre-trained non-equivariant baseline (with or without canonicalization). This evidences that test-time canonicalization is significantly more challenging in the 3D domain than in images (at least for molecular conformations with varying per-molecule rotations). Nevertheless, RECON is the

Table 4: EquiAdapt hyperparameter search and final test accuracy on the per-class rotated MNIST, FashionMNIST and GEOM-QM9 dataset variations.

| Learning rate | Prior weight | Test acc. (MNIST) | Test acc. (FashionMNIST) | Test acc. (GEOM-QM9) |
|---|---|---|---|---|
| 1e-4 | 0.5 | 89.05% | 62.22% | 40.12% |
| 1e-4 | 1.0 | 91.60% | 54.85% | 42.63% |
| 1e-4 | 10.0 | 93.83% | 70.30% | 49.41% |
| 1e-4 | 50.0 | 94.05% | 78.28% | 39.15% |
| 1e-4 | 100.0 | 92.97% | 79.00% | 46.11% |
| 3e-4 | 0.5 | 92.39% | 59.37% | 37.54% |
| 3e-4 | 1.0 | 92.41% | 65.35% | 43.82% |
| 3e-4 | 10.0 | 94.09% | 71.76% | 44.81% |
| 3e-4 | 50.0 | 94.67% | 75.96% | 46.75% |
| 3e-4 | 100.0 | 94.86% | 78.50% | 40.64% |
| 1e-3 | 0.5 | 92.17% | 54.75% | 43.22% |
| 1e-3 | 1.0 | 92.02% | 63.86% | 42.57% |
| 1e-3 | 10.0 | 94.76% | 71.92% | 48.17% |
| 1e-3 | 50.0 | 94.48% | 77.82% | 46.78% |
| 1e-3 | 100.0 | **95.00%** | 78.13% | 47.28% |
| 5e-3 | 0.5 | 92.47% | 54.61% | 45.33% |
| 5e-3 | 1.0 | 92.77% | 64.66% | 52.19% |
| 5e-3 | 10.0 | 93.82% | 72.13% | 47.50% |
| 5e-3 | 50.0 | 94.74% | 78.19% | **53.81%** |
| 5e-3 | 100.0 | 94.70% | **79.13%** | 48.72% |

Table 5: ESCNN hyperparameter search on MNIST and FashionMNIST. Test accuracy on the per-class rotated test sets.

| Emb. dim | Hidden dim | Learning rate | Test acc. (MNIST) | Test acc. (FashionMNIST) |
|---|---|---|---|---|
| 512 | 128 | $1e{-}3$ | 91.76% | - |
| 512 | 128 | $3e{-}4$ | 93.11% | 77.13% |
| 512 | 192 | $1e{-}3$ | 92.87% | 75.61% |
| 512 | 192 | $3e{-}4$ | 92.11% | 76.06% |
| 1024 | 128 | $1e{-}3$ | **94.59%** | **77.94%** |
| 1024 | 128 | $3e{-}4$ | 94.02% | 76.25% |
| 1024 | 192 | $1e{-}3$ | 93.99% | 73.52% |
| 1024 | 192 | $3e{-}4$ | 91.93% | 75.94% |

only test-time canonicalization that offers improvements over the baseline, with both EquiAdapt and IE-AE reducing performance over the pre-trained GCN. The SEGNN classifier is built using the SEGNN backbone as described in Appendix E.2.3, and was trained for 50 epochs with batch size 128 and learning rate $1e - 3$.

## G STATEMENTS

### G.1 USE OF LARGE LANGUAGE MODELS

Large language models were used to aid in writing (polishing text), retrieval of related work, generating code for plots, and implementing standard components.

### G.2 REPRODUCIBILITY STATEMENT

The experiments in this paper can be reproduced using the code provided in the repository at `https://github.com/ZIB-IOL/recon`.

### G.3 ETHICAL CONCERNS

Because our method relies on a data-driven approach to identify natural poses, it may be susceptible to dataset bias. Beyond this, we do not anticipate significant ethical concerns or negative societal impacts.

