# OpenReview forum: "RECON: Robust symmetry discovery via Explicit Canonical Orientation Normalization"
_ICLR.cc/2026/Conference — ICLR 2026 Poster_

### Official Review · Reviewer_XXPp · 2025-10-31

**Soundness:** 3
**Presentation:** 3
**Contribution:** 3
**Rating:** 6
**Confidence:** 3

**Summary:**

This paper seeks to build on IE-AEs by applying canonical orientation normalization to obtain a different equivalence class than with a typical IE-AE. The authors address instance-specific symmetries, such as rotations in images. The authors also present test-time canonicalization, which allows pre-trained models to be granted invariance with respect to a group action. The authors experiment with a couple of image benchmark datasets, as well as geometric graphs, with SO(2) and SO(3) symmetries.

**Strengths:**

1. The authors demonstrate the potential advantages of their approach over the most comparable baselines.

2. The presented method appears to be novel.

**Weaknesses:**

1. *The symmetries considered are limited.* The groups SO(2) and SO(3) do not constitute a diverse collection of symmetries, as lines 348-354 suggest.

2. *This is not the first work to consider 3-d symmetry,* contrary to the claim made in the abstract. In fact, this statement can mean one of two things: (1) the symmetry group is 3-dimensional (perhaps it is a Lie group); (2) the group action acts on a three-dimensional space. In either case, the claim made by the authors would be false. In fact, I do not believe that this work is the first to consider SO(3) symmetry: for example, see "Image to Icosahedral Projection for SO(3) Object Reasoning from Single-View Images" (Klee et al., 2022).

3. *There are a few key missing references.* In particular, the idea of using the Lie derivative to discover symmetry ("A Unified Framework to Enforce, Discover, and Promote Symmetry in Machine Learning" by Otto et al., 2023) and to subsequently use the discovered symmetries to construct an invariant feature space ("Symmetry Discovery Beyond Affine Transformations by Shaw et al., 2024) seems closely related to the present method and should be discussed, if not experimentally compared with. Another key reference in symmetry discovery is "Learning Infinitesimal Generators of Continuous Symmetries from Data" by Ko et al., 2924.

4. *More experimental comparison would be helpful.* I appreciate the experimental comparison which has been made, but it would be nice to see a comparison against a representative sample of other methods seeking to enforce invariance, such as an equivariant NN or else a method or two mentioned in weakness 3.

**Questions:**

1. Near line 150, the authors write: "In effect, we rely on the principle that structurally similar objects generally map to nearby points in Z." Is this really true in general? The authors mention empirical evidence, but what symmetries have been considered in these experiments? What does "nearby" mean in the G-invariant latent space? I think we need to be more precise about what this d_Z norm-induced distance is, or perhaps prove that *any* norm-induced distance suffices.

2. What kind of symmetries can you learn? (It seems like the bottle-neck could be the G-invariant encoder, which may be limited in the types of symmetries it can discover.) It seems only rotations are experimented with. The claim is that the method is quite general, but a claim that *any* symmetry can be discovered is quite doubtful, particularly in light of the lack of experimental evidence.

---

> ### Author Response · Authors · 2025-11-21
> **Response to reviewer XXPp**
>
> We thank the reviewer for their careful reading and constructive feedback. Below we clarify every point that was raised and describe the additional experiments we have now included. We provided a revised manuscript, with all changes highlighted in blue for ease of reference.
>
> ----
>
>
> > “The symmetries considered are limited. The groups SO(2) and SO(3) do not constitute a diverse collection of symmetries, as lines 348–354 suggest.”
>
> We agree that our experimental scope is limited to rotation groups. We have revised the corresponding line (L372) to tone down this statement and clarify that we study only rotation groups empirically. Our intention is to emphasize that RECON is in principle general enough to be applied on different data domains and groups, not to claim coverage of a broad variety of symmetry types experimentally. We acknowledge that certain symmetries or datasets might be challenging to discover with the current framework (see limitations).
>
> > “This is not the first work to consider 3-d symmetry, contrary to the claim made in the abstract…” (...) “I do not believe that this work is the first to consider SO(3) symmetry: for example, see "Image to Icosahedral…”
>
> Note that the provided reference from Klee et al [3] aims to find an equivariant architecture for SO(3)-type reasoning from single-view RGB images, built by mapping images onto an icosahedral representation and applying group convolutions there. It is not about symmetry discovery, but about designing an SO(3)-equivariant network in a certain setting.  However, you are correct: our 3D claim is inaccurate as is. Prior work such as LieGAN [1] or LaLiGAN [2] already discovers symmetries that include 3D rotation groups. For further clarification and context w.r.t the provided literature: the difference is that we estimate instance-level symmetry distributions over a (known) group G, while LieGAN and LaLieGAN provide symmetry discovery of the (unknown) group G at the dataset level. Additionally, our work features a canonicalization operator that is usable at test time with (any-architecture) pre-trained models. In contrast, the provided references do not directly expose a canonicalization function that provide benefits to downstream models at test-time (e.g. LieGAN discoveries can be leveraged for downstream use, but as part of a specialized EGNN architecture).
>
> With this nuance, and addressing the 3D concern, our contribution is about instance-level symmetry discovery in 3D groups. We removed the current claim and revised the abstract to make a precise, nuanced contribution statement. We thank you for the provided references, and added a discussion of LieGAN and LaLieGAN, as well as other relevant references in an updated Related work section. We also properly contextualized our 3D claim with additional references such as ImplicitPDF [5] and Alignist [6], which also learn distributions in SO(3) (although with caveats, which we discuss).
>
> > “There are a few key missing references. In particular, the idea of using the Lie derivative…”
>
> We appreciate the provided references and added them to the Related work section, and put them in context with our framework. However, note that these works solve a fundamentally different task, although they all are encompassed under the broad “symmetry discovery” label. In one way or another (e.g. through infinitesimal generators or reasoning with the Lie derivative), these works aim to infer a group G (and sometimes construct invariant features) from the data. In contrast, RECON assumes a known group, and focuses on learning instance-level symmetry distributions over the group, alongside a canonicalization operator that can be placed in front of any downstream model. Therefore, the aforementioned approaches are very much complementary: they target symmetry discovery at the level of inferring a dataset-level (unknown) G, while our method provides a fine-grained discovery of an instance-level distribution over G (known). Given the differences in problem setup, we do not incorporate a quantitative comparison in the current revision, but we agree that bridging or combining these lines of work is a very interesting research direction. We would like to add that while many methods indeed explore discovering the group, RECON addresses a less explored regime (discovering instance-level symmetry distributions over a prescribed group), together with a plug-in canonicalization that has the potential of broad adoption. While approaches such as Alignist [7] can also model distributions on SO(3), they are not as general (symmetry and data type wise) and require CAD priors or some form or supervision (we discuss this on the updated related work), whereas RECON operates directly on unlabeled data through the class-pose encoder. We thank you for providing valuable references that give more context to the benefits and limitations of our method w.r.t. the current literature.

---

> > ### Comment · Reviewer_XXPp · 2025-11-24
> >
> > I appreciate the response by the authors, and many of my initial concerns are resolved.
> >
> > I am not yet convinced that the distinction between methods such as LieGAN and the method presented in this paper is entirely correct. One possible meaning of instance-specific symmetry is illustrated as follows: for an image of a perfectly-rounded "O", discover the rotational symmetry, which results in the statement that the specific image is unchanged when rotated. (And the instance-specific symmetry is in contrast to an image of a perfect square, which is symmetric only under rotations by multiples of 90 degrees.) However, it seems this is not the case the authors are dealing with, as evidenced by the experiment with MNIST and FashionMNIST. It seems that the authors are, for each training image, constructing a collection of transformed images based on a given group action: however, the specific transformations chosen can vary per image (or per image class). The task is to recover the distribution over the prescribed group that is specific to each collection of transformed images (or class of images), if I understand correctly. But this problem just sounds like parallel "micro" LieGAN problems, or else it is similar to the LieGAN experiment of discovering rotational symmetry from partial trajectories.
> >
> > The production of a distribution over a prescribed group G is similar to discovering a symmetry subgroup of a prescribed group G, which is a problem LieGAN and similar methods handle: these methods don't discover an entirely novel group G, but rather a subset of elements of prescribed/known group under which data is symmetric. Thus, I think the distinction between the so-called "instance-specific" symmetry discovery in the present paper and the symmetry discovery of methods such as LieGAN is not as clear as the authors claim.

---

> ### Author Response · Authors · 2025-11-21
>
> > “More experimental comparison would be helpful… it would be nice to see a comparison against a representative sample of other methods seeking to enforce invariance, such as an equivariant NN or else a method or two mentioned in weakness 3.”
>
> We agree that broader quantitative comparisons would be valuable. We are working on additional comparisons for the camera-ready version, in particular with Equivariant Adaptation (EA) as another test-time canonicalization method, and with at least one representative equivariance-learning architecture (e.g., Partial G-CNNs [5]).
>
> References:
>
> [1] Generative Adversarial Symmetry Discovery.
>
> [2] Latent Space Symmetry Discovery.
>
> [3] Image to Icosahedral Projection for SO(3) Object Reasoning from Single-View Images. Klee et al (2022)
>
> [4] Equivariant Adaptation of Large Pretrained Models
>
> [5] Learning Partial Equivariances from Data. Romero, David W., and Suhas Lohit.
>
> [6] Murphy, K., Esteves, C., Jampani, V., Ramalingam, S., & Makadia, A. (2022). Implicit-PDF: Non-Parametric Representation of Probability Distributions on the Rotation Manifold.
>
> [7] Vutukur, S. R., Haugaard, R. L., Huang, J., Busam, B., & Birdal, T. (2024). Alignist: CAD-Informed Orientation Distribution Estimation by Fusing Shape and Correspondences.
>
> ----
>
> We thank you for the time invested, insightful questions and provided references, which we believe strengthen the manuscript. In the new revision (changes highlighted in blue), we have included other interesting analyses on equivalence class construction, neighborhood size sensitivity, and computational overhead, as well as some updated results. We hope our latest revision and responses clarify your questions and concerns.
>
> Please let us know if you have any follow-up / additional questions.
>
> Best regards,
>
> The Authors

---

> ### Author Response · Authors · 2025-11-27
> **Response to reviewer XXPp**
>
> We thank you for the follow-up and for the concerns raised.
>
> > “One possible meaning of instance-specific symmetry is illustrated as follows:”
>
> You are correct that the term “instance-specific symmetry” could (rightfully) be interpreted as “the stabilizer $S_x =${ $g\in G | gx = x$ } of each instance”, like in the example you provided, in which you speak about the stabilizer of a perfect “O” (which is $S_{circle} = SO(2)$) vs. stabilizer of a perfect square (which is $S_{square} =$90º rotations). As you point out, this is not the task we consider. Our notion of “instance-level symmetry distribution” concerns the discovery of the orbit/pose distribution in the data (orbit being $G_x =$ {$g x | g \in G$ }), rather than characterizing the stabilizer $S_x$. That is, we learn (for each instance) a probability distribution over group elements whose action populates the part of the orbit $G_x$ that is actually realized in the data. Given a prescribed group $G$, we learn such a distribution and use it to define a data-aligned canonical representative.
>
> We agree that the use of the word “symmetry” here is overloaded and demands clarification, and in the updated, revised version we consistently rephrase this as “instance-level pose/orbit distributions” over a prescribed group, and explicitly distinguish this from discovering input-dependent stabilizers in the sense of your “perfect O vs square” example.
>
> At the same time, note how the two notions are closely related: if an input $x$ has a non-trivial stabilizer $S_x =${ $g\in G | gx = x$ } , then many group elements act indistinguishably on $x$ (i.e., $g$ and $g \cdot s$ with $s \in S_x$ produce the same data point). As a consequence, the pose distribution is only identifiable modulo $S_x$ and tends to become broad or multimodal along those symmetry directions. This can be problematic, since in that regime the canonical pose is inherently ambiguous and naive canonicalization can yield out-of-distribution representatives (Figure 4), but our robust canonicalization objective is precisely designed to yield in-distribution canonical representatives despite this ambiguity.
>
> For empirical evidence, consider the digit ''8'' in MNIST. Although in practice, left-to-right handwriting direction often disambiguates whether an ''8'' is upside down or upright, for many input 8s, these cues are effectively absent, and the two poses are practically indistinguishable. Consequently, a non-negligible fraction of 8s is mapped to an upside-down configuration. This is visible in the histograms of Figure 13 for class 8: there is a clear non-zero mass outside the $[-90^\circ, 90^\circ]$ regime, caused by these highly symmetric (rotated) 8s, which induce additional density near $180^\circ$. This contrasts with digits such as 3, 4, or 7, which have trivial stabilizers (i.e., do not exhibit such stabilizer-induced ambiguity), and therefore yield sharply concentrated histograms. Nevertheless, our Tukey-Fréchet canonicalization objective is designed to handle precisely this situation: even in an idealized worst-case scenario where all 8s are perfectly symmetric and the pose distribution is exactly bimodal (i.e. probability 0.5 on upright and 0.5 on upside-down), the optimization does not average the two modes into an unstable ``in-between'', out-of-distribution pose, but instead admits minimizers aligned with one of the symmetric modes as canonical in-distribution solutions, enabling stable test-time canonicalization (we discussed this in Appendix C). In other words, our method is robust to these non-trivial stabilizers.
>
> We have updated the use of “pose distribution” on the phrasing across all sections in the manuscript, and added a short discussion on the stabilizer vs orbit nuance in Appendix D.1.

---

> ### Author Response · Authors · 2025-11-27
> **Response to reviewer XXPp**
>
> > “But this problem just sounds like parallel "micro" LieGAN problems” (...) “the distinction between the so-called "instance-specific" symmetry discovery in the present paper and the symmetry discovery of methods such as LieGAN is not as clear”
>
> We agree that both LieGAN and our method assume a prescribed search space $G$ and deal with distributions over group elements, so it is indeed important to be precise about what differs.
>
> LieGAN is trained to find \emph{global} transformation directions that leave the data distribution approximately invariant. The identity transformation is always a trivial solution, but LieGAN is regularized to deviate from the identity, so as to search for non-trivial directions in the Lie algebra along which the discriminator’s loss stays low. For instance, consider the experiment they performed on MNIST with uniform $\pm 45^\circ$ rotations (see their Appendix C.4). In here, the only rotation that leaves all pose histograms invariant is actually the identity. Nevertheless, in practice, LieGAN’s regularization still tries to find a (typically low-dimensional) set of directions in transformation space that keeps the marginal $p(x)$ as invariant as possible, and therefore in this dataset, they recover a dominant $\mathrm{SE}(2)$-type direction that keeps the empirical $p(x)$ nearly invariant and speaks about rotational symmetries being present in the data (with some degree of accuracy): “The discovery result (...) can be interpreted as a mixture of rotation (L[1, 0] = −L[0, 1] = 0.66) and translation (L[0, 2] = 0.08), where the magnitude of rotations is larger than the magnitude of translations” [1]. This estimation is describing the \emph{global} symmetry directions along which the distribution can move with little change and, through the learned coefficient distribution, can also capture aspects of the magnitude of the transformations at the dataset level, but it does not model per-instance pose distributions explicitly.
>
> We differ in both the objective (and hence the estimations we make) and the granularity. For a prescribed $G$, RECON learns \emph{instance-dependent} pose/orbit distributions $\mu_x$ on $G$ together with a canonicalization map $C(x)$. These $\mu_x$ are not required to preserve $p_d$; they are meant to describe, for each $x$, which group elements actually occur in the data along its orbit (and with what frequencies). It is related but different from LieGAN’s symmetry discovery objective, and provides a fine-grained, instance-level description of the magnitude and shape of the transformations present in the data. Moreover, the distributions we discover support a gauge-fixing operation that “breaks” or “removes” symmetry by mapping to in-distribution canonical representatives for test-time canonicalization (in contrast to “preserving” symmetries, which is at the heart of LieGAN’s objective).
>
> Regarding the “parallel micro LieGAN problems” analogy, there is indeed a conceptual connection (since this “parallel, per-orbit LieGAN” idea would indeed discover per-orbit transformation directions in which the orbits are kept approximately invariant). However, in its standard formulation, LieGAN uses a single global distribution over group elements, independent of $x$, optimized to keep $p_d$ approximately invariant, whereas RECON is explicitly parameterized to output $\mu_x$ and $C(x)$, allowing different orbits to have different pose distributions and enabling canonicalization. A hypothetical orbit-wise LieGAN method would be solving a task similar in spirit to ours, but would still differ in both objective  (distribution preservation vs modeling pose distribution) and estimations, and in how the learned distributions are used (in our case, for a group invariant operation through test-time canonicalization).
>
> We corrected our manuscript to show that our distinction is about the granularity of the estimations and the way the learned distributions are learned and used downstream.
>
> References:
>
> [1] Generative Adversarial Symmetry Discovery

---

> ### Author Response · Authors · 2025-11-27
> **Update to reviewer XXPp**
>
> ----
>
>
> We thank you for the raised concerns raised regarding the nuances about the symmetry discovery task and the comparison with LieGAN.
>
> To address your request for broader experimental comparisons, we have updated our manuscript and added a set of experiments comparing RECON to other test-time canonicalization methods (EquiAdapt [1]), as well as against fully equivariant (SE(2) equivariant Steerable CNNs [2] / SEGNN [3]) and learned equivariance (Partial G-CNNs [4]) baselines.
>
> Overall, we find that RECON’s canonicalization provides competitive (MNIST) or better performance  than EquiAdapt (FashionMNIST, GEOM-QM9), while being trained without supervision (as opposed to EquiAdapt’s). In the imaging experiments, our canonicalization consistently outperforms Partial G-CNNs (sometimes by a large margin as in FashionMNIST). Against fully equivariant baselines, RECON comes moderately close in performance (MNIST) and even surpasses them (FashionMNIST), while retaining the benefits of being a plug-and-play layer that can be attached in front of an arbitrary backbone. On GEOM-QM9, we underperform fully equivariant networks, but still are able to improve over the pre-trained baseline, IE-AE and EquiAdapt.
>
>
> ---
>
> We hope this addresses the raised concerns and thank you for providing valuable feedback. A brief global summary of all changes across revisions is also provided in our general official comment. Please let us know if you have any follow-up / additional questions.
>
> Best regards,
>
> The Authors
>
> ---
>
> [1] Mondal, A. K., Panigrahi, S. S., Kaba, S.-O., Rajeswar, S., & Ravanbakhsh, S. (2023). Equivariant Adaptation of Large Pretrained Models
>
> [2] Cesa, G., Lang, L., & Weiler, M. (2022). A Program to Build E(N)-Equivariant Steerable CNNs.
>
> [3] Brandstetter, J., Hesselink, R., Pol, E. van der, Bekkers, E. J., & Welling, M. (2022). Geometric and Physical Quantities Improve E(3) Equivariant Message Passing
>
> [4] Learning Partial Equivariances from Data. Romero, David W., and Suhas Lohit.

---

### Official Review · Reviewer_BHNi · 2025-10-31

**Soundness:** 3
**Presentation:** 3
**Contribution:** 3
**Rating:** 6
**Confidence:** 3

**Summary:**

The paper introduces a method for unsupervised discovery of instance-specific symmetry distributions in data. Building upon class–pose decomposition frameworks (like Invariant-Equivariant Autoencoders), it corrects arbitrary canonical poses by estimating a Fréchet mean offset and applying a canonical orientation normalization. This yields consistent, data-aligned canonical representations and enables applications such as OOD pose detection and test-time canonicalization. The method is validated on both 2D datasets (MNIST, FashionMNIST) and 3D molecular datasets (GEOM-QM9).

**Strengths:**

- The canonical orientation normalization via the Fréchet mean is simple yet theoretically grounded and effective for unsupervised symmetry discovery.

- The paper provides rigorous proofs and clear geometric intuition, which is scalable to both SE(2) for imaging and SO(3) for geometric graphs.

- This method can be applied to any invariant-equivariant backbone, making it broadly useful.

- The test-time canonicalization and OOD detection tasks are compelling, showing both research and applied potential.

**Weaknesses:**

- Lack of real-world validation in the experiments. Most experiments involve synthetic rotations or clean 3D datasets; robustness under complex, cluttered scenes for real-world objects, for example, objects in 2D images in the wild, like COCO/ImageNet, or scanned 3D objects in OmniObject3D/GSO.

- The sensitivity to hyperparameters (e.g., k-nearest neighbors in class construction) could be explored more systematically.

- Lacks direct quantitative comparisons with very recent equivariance-learning approaches (e.g., Partial G-CNNs, VP-GCNNs) beyond conceptual discussion.

**Questions:**

- What is the training overhead and inference time for RECON’s symmetry detection/OOD pose detection/ test-time canonicalization/reconstruction?

- There are many learning-based methods that take take 3D object model as input and model symmetry pattern/distribution in SO(3), e.g., Implicit-PDF[1], Alignist[2]. Could you explain if RECON can be extended to a more generalized setup like the above-mentioned methods, and what would be the main advantage of RECON compared to those methods if the same datasets apply (for example, Symsol)?

References:

[1] Murphy, K.A., Esteves, C., Jampani, V., Ramalingam, S., Makadia, A.: Implicitpdf: Non-parametric representation of probability distributions on the rotation manifold. In: Proceedings of the 38th International Conference on Machine Learning. pp. 7882–7893 (2021).

[2] Vutukur, Shishir Reddy, Rasmus Laurvig Haugaard, Junwen Huang, Benjamin Busam, and Tolga Birdal. "Alignist: CAD-Informed Orientation Distribution Estimation by Fusing Shape and Correspondences." In European Conference on Computer Vision, pp. 351-369. Cham: Springer Nature Switzerland, 2024.

---

> ### Author Response · Authors · 2025-11-21
> **Response to reviewer BHNi**
>
> We thank the reviewer for their careful reading and constructive feedback. Below we clarify every point that was raised and describe the additional experiments we have now included. We provided a revised manuscript, with all changes highlighted in blue for ease of reference.
>
> ----
>
> > “Lack of real-world validation in the experiments”...“robustness under complex, cluttered scenes...”
>
> We agree that our current image experiments do not cover cluttered, in-the-wild scenes such as COCO/ImageNet-style data. This stems from the instance-centric nature of the class-pose method, which we discuss in the limitations section, and frame object-level discovery as future work. We also would like to highlight that GEOM-QM9 is a real-world molecular dataset rather than a synthetic one: each datapoint is a distinct, chemically realistic conformation with unique 3D coordinates (RMSD up to 1.5Å across conformers of the same molecule). This gives evidence that RECON can handle non-trivial, noisy geometric variability beyond perfectly clean synthetic data.
>
>
> > “The sensitivity to hyperparameters (e.g., k-nearest neighbors in class construction) could be explored more systematically.”
>
> We ran the RECON pipeline (Algorithm 1 + training of $\Theta$ and $\Lambda$) with multiple seeds and now report mean and standard deviation for symmetry discovery, OOD detection and test-time canonicalization (please refer to the updated values in the manuscript and to the new Figure 9). In the latter, we empirically show that symmetry discovery performance varies mildly across a reasonable range of k for constructing the equivalence class, indicating that the task is not overly sensitive to the neighborhood size. We also added a deeper analysis on the neighbors' impact on prediction error and class equivalence construction (see Appendix D.1, D.2), updated some of our neighbor size choices and results, and provided several insights. Lastly, we explored failure modes of our equivalence class and provided both quantitative and qualitative results, making the role of k (and its limitations) more transparent.
>
> > “Lacks direct quantitative comparisons with very recent equivariance-learning approaches (e.g., Partial G-CNNs, VP-GCNNs)”
>
> We agree that broader quantitative comparisons to recent equivariant architectures would be valuable. We are also exploring additional comparisons for the camera ready version, particularly with Equivariant Adaptation (EA) [3], as it is another test-time canonicalization method, and will include it alongside an equivariance-learning method comparison such as Partial G-CNNs for the camera-ready version.
>
> > “What is the training overhead and inference time for RECON’s symmetry…”
>
> We report the additional training cost of learning $\Lambda$ and $\Theta$ in Appendix E. For clarity, this is the only extra training introduced by RECON: the class-pose backbone is trained independently, and our method is applied on top of a pre-trained class-pose model. We have also added a computational analysis section (Appendix D.4) with k-NN wall-clock times and measurements of the inference overhead. The k-NN search is a one-time computation used only to create pseudo-labels (Algorithm 1) for training $\Theta$ and $\Lambda$. Overall, the added cost from RECON is small. Test-time canonicalization is implemented as a single, light-weight module, and the performance gains reported in Figure 6d come with only a modest increase in inference time.
>
> > “There are many learning-based methods that take take 3D object model as input and model symmetry pattern/distribution in SO(3), e.g., Implicit-PDF[1], Alignist[2]”...“Could you explain if RECON can be extended to a more generalized setup like the above-mentioned methods…”
>
> We thank you for the provided relevant references. Both methods indeed model probability distributions over the group. In particular, Implicit-PDF [1] models instance-level distributions over SO(3) with an implicit density and pose supervision, and Alignist [2] estimates orientation distributions using CAD shape priors and correspondence distributions. Our method, in contrast, requires neither pose labels nor CAD models and operates on general groups supported by the class-pose backbone. We updated the related work section and added a discussion with both these methods, as well as other missing relevant literature.
>
> References
>
> [1] Murphy et al., “Implicit-PDF: Non-parametric Representation of Probability Distributions on the Rotation Manifold”, ICML 2021.
>
> [2] Vutukur et al., “Alignist: CAD-Informed Orientation Distribution Estimation by Fusing Shape and Correspondences”, ECCV 2024.
>
> [3] Equivariant Adaptation of Large Pretrained Models
>
> ----
>
> We thank you for the time invested, insightful questions and provided references, which we believe strengthen the manuscript. We hope our latest revision clarifies your questions and concerns. Please let us know if you have any follow-up / additional questions.
>
> Best regards,
> The Authors

---

> ### Author Response · Authors · 2025-11-27
> **Update to reviewer BHNi**
>
> We thank you again for your detailed feedback. We have updated our manuscript and added a set of experiments quantitatively comparing RECON to other test-time canonicalization methods (EquiAdapt [1]), as well as against fully equivariant (SE(2) Equivariant Steerable CNNs [2] / SEGNN [3]) and learned equivariance (Partial G-CNNs [4]) baselines.
>
> ---
>
> Overall, we find that RECON’s canonicalization provides competitive (MNIST) or better performance  than EquiAdapt (FashionMNIST, GEOM-QM9), while being trained without supervision (as opposed to EquiAdapt’s). In the imaging experiments, our canonicalization consistently outperforms Partial G-CNNs (sometimes by a large margin as in FashionMNIST). Against SE(2) Equivariant Steerable CNNs, RECON comes moderately close in performance (MNIST) and even surpasses them (FashionMNIST), while retaining the benefits of being a plug-and-play layer that can be attached in front of an arbitrary backbone. On GEOM-QM9, we underperform fully equivariant networks, but still are able to improve over the pre-trained baseline, IE-AE and EquiAdapt.
>
> ---
>
> A brief global summary of all changes across revisions is also provided in our general official comment. We hope this strengthens your views about RECON and thank you for providing valuable feedback towards improving our manuscript. Please let us know if you have any follow-up / additional questions.
>
> Best regards,
>
> The Authors
>
> ---
>
> References
>
> [1] Mondal, A. K., Panigrahi, S. S., Kaba, S.-O., Rajeswar, S., & Ravanbakhsh, S. (2023). Equivariant Adaptation of Large Pretrained Models
>
> [2] Cesa, G., Lang, L., & Weiler, M. (2022). A Program to Build E(N)-Equivariant Steerable CNNs.
>
> [3] Brandstetter, J., Hesselink, R., Pol, E. van der, Bekkers, E. J., & Welling, M. (2022). Geometric and Physical Quantities Improve E(3) Equivariant Message Passing
>
> [4] Learning Partial Equivariances from Data. Romero, David W., and Suhas Lohit.

---

### Official Review · Reviewer_ff67 · 2025-11-01

**Soundness:** 2
**Presentation:** 3
**Contribution:** 2
**Rating:** 4
**Confidence:** 3

**Summary:**

The paper introduces RECON, an unsupervised framework for discovering instance-specific symmetries from data.
RECON builds upon class-pose decomposition methods (e.g., IE-AE) and addresses the key limitation of arbitrary canonical poses by normalizing them into natural orientations through an estimated Tukey-Fréchet mean of relative transformations.
This canonical orientation normalization yields interpretable, data-aligned symmetry distributions and enables downstream applications such as (i) OOD pose detection and (ii) test-time canonicalization that grants invariance to pretrained models without retraining.
Experiments on 2D image datasets (MNIST, FashionMNIST) and 3D molecular data (GEOM-QM9) demonstrate the effectiveness of the proposed symmetry discovery method itself as well as its benefits in downstream tasks.

**Strengths:**

- The approach discovers symmetry distributions from unlabeled data and provides meaningful, identity-centered canonicalizations that align with intuitive notions of natural pose.

- Interesting and elegant idea: The proposed Tukey-Fréchet-mean–based canonical orientation normalization provides a simple yet effective solution to the issue of arbitrary canonicalization in class-pose methods. The modeling is conceptually clear, mathematically grounded, and intuitively appealing.

- RECON can grant group invariance to pretrained models without retraining, functioning as a plug-in canonicalization step that applies irrespective of model architecture. This design significantly enhances the method’s usability and potential impact.

- The paper is clearly structured and communicates its ideas effectively, making both the theoretical formulation and practical implications easy to understand.

**Weaknesses:**

- Inaccurate contribution claim. The paper claims to “-for the first time– extend symmetry discovery to 3D groups,” but prior works such as [1] and [2] have already explored 3D group symmetries.

- The current method is restricted to known transformation groups, primarily rotations (SO(2) and SO(3)), limiting its applicability to more general groups or unknown symmetry settings where the underlying group structure must be inferred.

- The method assumes that variations within each equivalence class are small and can be cleanly separated from group-induced transformations. It remains unclear how reasonable this assumption is in more complex, natural data.

- Computational cost and scalability: The reliance on k-nearest-neighbor search in the invariant space for each input (Algorithm 1) may be computationally expensive for large-scale or high-dimensional datasets.
The paper does not discuss computational cost or whether neighbor computations can be cached or reused during training and inference.

- Some key quantitative results (e.g., Figure 6) appear to be based on single runs without repeated trials or statistical analysis, making it difficult to assess robustness and reliability.
Moreover, the results in Figure 6(d) show a noticeable gap from the so-called upper bound in Table 1.

**Questions:**

- The equivalence class construction plays a crucial role in the proposed pipeline. Could the authors also provide hit rate results in the image domain?

- Could the authors further discuss the practical utility of discovering symmetry distributions?
It seems that the most tangible application demonstrated is test-time canonicalization, yet other works such as [3] and [4] can achieve a similar effect given a known group structure, without explicitly estimating instance-level distributions.

- In the implementation, does each equivalence class correspond to the entire semantic class, or is it constructed per-sample with a fixed number of neighbors (e.g., 25 as mentioned)?

- How would the method perform on real-world datasets where each object appears at a unique orientation and images are not repeated across poses, unlike the synthetic settings used in this paper?

Reference:

[1] Generative Adversarial Symmetry Discovery.

[2] Latent Space Symmetry Discovery.

[3] Affine Steerable Equivariant Layer for Canonicalization of Neural Networks.

[4] Equivariant Adaptation of Large Pretrained Models.

---

> ### Author Response · Authors · 2025-11-21
> **Response to reviewer ff67**
>
> We thank the reviewer for their careful reading and constructive feedback. Below we clarify every point that was raised and describe the additional experiments we have now included. We provided a revised manuscript, with all changes highlighted in blue for ease of reference.
>
> ----
>
> > “Inaccurate contribution claim. The paper claims to “-for the first time–...”
>
> Thanks for pointing this out; this is correct, our claim is inaccurate as is. Prior work such as LieGAN [1] and LaLiGAN [2] already discovers symmetries that include 3D rotation groups. For further clarification and context w.r.t the provided literature: the difference is that we estimate instance-level symmetry distributions over a (known) group G, while LieGAN and LaLieGAN provide symmetry discovery of the (unknown) group G at the dataset level. Additionally, our work features a canonicalization operator that is usable at test time with (any-architecture) pre-trained models. In contrast, the provided references do not directly expose a canonicalization function that provide benefits to downstream models at test-time (e.g. LieGAN discoveries can be leveraged for downstream use, but as part of a specialized EGNN architecture).
>
> With this nuance, and addressing the 3D concern, our contribution is about instance-level symmetry discovery in 3D groups. We removed the current claim and revised the abstract to make a precise, nuanced contribution statement. We thank you for the provided references, and added a discussion of LieGAN and LaLieGAN, as well as other relevant references in an updated Related work section. We also properly contextualized our 3D claim with additional references such as ImplicitPDF [5] and Alignist [6], which also learn 3D distributions (although with caveats, which we discuss).
>
> > “The current method is restricted to known transformation groups…”
>
> We agree this is a limitation; RECON discovers distributions over a pre-defined group G, and can not infer group-agnostic transformations. We discussed this explicitly in the limitations paragraph and framed discovering group-agnostic transformations as future work. At the same time, there are methods that do target this goal of discovering a (dataset-level) group (e.g., LieGAN), although they do not learn instance-level distributions nor a test-time canonicalization function. In this sense, RECON and this class of group discovery methods are complementary methods, all under the label of “symmetry discovery”. RECON addresses a different, we believe less explored angle in symmetry discovery: it assumes a known G and focuses on discovering instance-level distributions over the group, and also provides extra model-agnostic applications with strong applicability potential. We see these symmetry discovery regimes as complementary rather than competing, and agree that combining both is a natural and interesting direction, although it is beyond the scope of the current work.

---

> ### Author Response · Authors · 2025-11-21
> **Response to reviewer ff67**
>
> > “The method assumes that variations within each equivalence class are small…” (...) “The equivalence class construction plays a crucial role in the proposed pipeline. Could the authors also provide hit rate results in the image domain?”
>
> We updated the manuscript and added a deeper analysis on the neighbors' impact on prediction error and hit rate (see Appendix D.1, D.2), and updated some of our neighbor size choices and results. We provide several insights that we hope address your concerns.
>
> Firstly, we empirically show that symmetry discovery performance varies mildly across a reasonable range of k for constructing the equivalence class, indicating that the task is not overly sensitive to the neighborhood size. Imaging tasks tend to favor smaller neighborhood sizes which provide higher hit rates, although we find that too few neighbors result in inaccurate symmetry parameter estimation. In effect, k=5 achieves the highest hit rate, yet k=10 yields the lowest prediction error. For GEOM-QM9, we found k=25 to work better, where smaller k (k=10) spiking errors in some molecules. We added new figures (Figure 9, Figure 12) and updated existing ones in the manuscript to reflect the latest, slightly better metrics obtained after the last change (k=10 now in images, instead of k=25 before).
>
> We now answer the question: “The method assumes that variations within each equivalence class are small…” . FashionMNIST dataset shows a relatively lower (~0.84) average hit rate (Figure 12c). We show through quantitative and qualitative analyses that this arises predominantly from inter-class feature overlap in the latent space from some inputs (e.g., between some "sneaker" and "ankle boot" classes which share lots of features), rather than from the magnitude of intra-class variation. We provide metrics and visualizations of this failure mode in the updated manuscript (Appendix D.2, Figure 11). Our more in-depth class equivalence analysis shows that our method thrives in latent spaces with clear separation between classes, and is not required to have small within-class variations in the data. Algorithm 1 only relies on local neighborhoods in the invariant space: for each $x$, it uses its k nearest neighbors in $Z$ to approximate $[x]$. In practice, this means RECON performs well whenever, for most points, a majority of their nearest neighbors share the same semantic class, even if the global diameter of that class in Z is large. This is consistent with our formal equivalence relation, which groups points into $\varepsilon$-connected components in Z: what matters is connectivity and separation from other classes, not small intra-class diameter. As an empirical example, GEOM-QM9 exhibits non-trivial intra-class conformational variations (RMSD ≤1.5Å, see visualizations in Figure 6) yet achieves a 0.95 hit rate. This is because molecular identity has a very strong and clear signal from compositional and topological invariants e.g. number of atoms of X element, that remain perfectly constant across conformers, which allows the SEGNN encoder to more easily separate conformers of different molecules. Our equivalence class definition behaves well as a result.
>
> When inter-class overlap does occur (as in FashionMNIST, see Figure 10 for an overview on the neighbor class overlap in the low hit rate regime), the remedy is to increase class separation in the latent space, not to reduce intra-class variation. Conveniently, this is a well-researched problem and established techniques like contrastive pretraining or self-supervised objectives excel at sharpening class boundaries without supervision (e.g., DINO, BYOL). These objectives are perfectly compatible with our framework.
>
> We thank you for the question and have updated our manuscript accordingly with a more thorough analysis on the equivalence class failure modes and mitigation strategies.
>
>
> > “Computational cost and scalability: The reliance on k-nearest-neighbor…”
>
> We updated our manuscript with a computational analysis section (Appendix D.4) with k-NN wall-clock times and additional measuring of the inference overhead. Note that the k-NN search is a one-time computation used only to create pseudo-labels (Algorithm 1) for downstream training of $\Theta$ and $\Lambda$. Furthermore, this computation admits batch processing and its complexity can be improved using highly optimized, publicly available implementations. Altogether, the use of k-NN in Algorithm 1 does not constitute a bottleneck to scaling to large dataset sizes.

---

> ### Author Response · Authors · 2025-11-21
> **Response to reviewer ff67**
>
> >  “Could the authors further discuss the practical utility of discovering symmetry distributions? It seems that…”
>
> For the sole goal of test-time canonicalization, both Equivariance Adaptation (EA) [3] and EquivarLayer [4] can indeed grant invariance to a known group without modeling per-instance symmetry distributions, much like our plug-in canonicalization step. However, we argue that our method has additional, broader practical advantages.
>
> Firstly, canonicalization-wise; Equivariant Adaptation (EA) [3] places an equivariant canonicalizer before a frozen predictor, and trains only the canonicalizer using a dataset-dependent canonicalization prior (what they coin the “zero-shot” setup); for smaller classifiers they also propose joint fine-tuning of the predictor (coined “fine-tune” setup). In both cases, the canonicalization function is tied to a specific pretrained model, its size, and the task during training. In contrast, our canonicalizer is trained independently of any downstream predictor and can be plugged in front of any model that operates on the same input domain. This decoupling avoids retraining or task-specific adaptation of the canonicalizer whenever the downstream model changes. Affine Steerable EquivarLayer [4] learns a canonicalizer trained independently of the pre-trained model, but we note that it is trained using known random transforms i.e. pose labels. Our canonicalization, however, is obtained without any supervision.
>
> Our second argument concerns the distributional output we obtain, as opposed to simply learning a canonicalization function as in [3,4]. This distributional output already supports additional downstream tasks beyond test-time canonicalization in our experiments, such as OOD pose detection. In addition, the learned distributions can naturally provide information about pose ambiguity or multi-modality, enable quantification of symmetry patterns in a dataset (e.g., by clustering the predicted parameters $\Theta(x)$), or be used to generate geometry-aligned augmentations by sampling from the learned symmetry distribution.
>
> Lastly, RECON is agnostic to the particular class-pose backbone, which offers one last practical advantage. It can sit on top of any method that provides an invariant latent space $\eta(x)$ and a pose map $\psi(x)$. This modularity means that advances in class-pose architectures can be directly leveraged to improve symmetry discovery, OOD detection, and test-time canonicalization within our framework. By contrast, EA and EquivarLayer are instantiated as specific canonicalization architectures rather than generic wrappers around arbitrary class-pose backbones, therefore they do not admit this form of modularity.
>
> We thank you for providing valuable references that give more context to the benefits and limitations of our method w.r.t. the current literature. We have included the provided references (and other relevant ones) in the updated Related work section.
>
> > “In the implementation, does each equivalence class correspond to the entire semantic class, or is it constructed per-sample with a fixed number of neighbors (e.g., 25 as mentioned)?”
>
> It is indeed computed per-sample; using the class information would go against the supervision-free spirit of the work.
>
> > “Some key quantitative results (e.g., Figure 6) appear to be based on single runs without repeated trials…”
>
> We ran the RECON pipeline (Algorithm 1 + training of $\Theta$ and $\Lambda$) with multiple seeds and now report mean and standard deviation for symmetry discovery, OOD detection and test-time canonicalization (please refer to the updated values in the manuscript and Figure 9). Overall, the downstream task trends remain the same: our method consistently improves over the baselines. We have updated our manuscript accordingly.
>
> > “How would the method perform on real-world datasets where each object appears at a unique orientation and images are not repeated across poses…”
>
> RECON does not require repeated views of the same object; it relies on semantic similarity and proximity in the latent space. Our GEOM-QM9 experiment can serve as stronger evidence: each datapoint is a real-world molecular conformation with a strictly distinct 3D geometry (unique atomic coordinates, RMSD up to 1.5Å) that never repeats, yet RECON achieves a 0.95 hit rate and accurate symmetry discovery, because all conformers of the same molecule share invariant compositional structure (atom types, connectivity) which maps them close in the latent space. The same holds for MNIST/FashionMNIST, where each handwritten digit is unique; however, variations here are certainly more trivial and one could possibly consider many inputs practically identical. We emphasize that the biggest limitation of our method is not unique orientations, but object-centric vs instance-level discovery, which we discussed in the limitations section and frame as promising future work.

---

> ### Author Response · Authors · 2025-11-21
> **Response to reviewer ff67**
>
> > “Moreover, the results in Figure 6(d) show a noticeable gap from the so-called upper bound in Table 1.”
>
> The “upper bound” is essentially the oracle case that exactly inverts test transforms (i.e., perfect canonicalization). RECON is a noticeably hard setting with no access to those transforms and no retraining of the downstream model, so a gap is expected. In images the gap is modest, $\approx$8 percentage points (pp) on MNIST and $\approx$9 pp on FashionMNIST, particularly when compared with ResNet18 ($\approx$23-25 pp gap). This is largely explained by imperfect canonicalization on inputs whose invariant embeddings have lower hit-rate or inter-class overlap, as we show in the updated equivalence class analysis in Appendix D.2. In 3D, the gap is larger ($\approx$40 pp on GEOM-QM9). We believe this is because we re-compute O(3) features after canonicalization which serve as input features to the GCN, and tiny pose errors get amplified when recomputing geometric features after rotation. Nevertheless, our canonicalization improves over the baseline, and in 3D data, it presents possibly one of the few alternatives alongside Equivariant Adaptation to learning a test-time canonicalization function in 3D data. We are working on a comparison against EA in test-time canonicalization for the camera-ready version.
>
> References:
>
> [1] Generative Adversarial Symmetry Discovery.
>
> [2] Latent Space Symmetry Discovery.
>
> [3] Affine Steerable Equivariant Layer for Canonicalization of Neural Networks.
>
> [4] Equivariant Adaptation of Large Pretrained Models
>
> [5] Murphy, K., Esteves, C., Jampani, V., Ramalingam, S., & Makadia, A. (2022). Implicit-PDF: Non-Parametric Representation of Probability Distributions on the Rotation Manifold.
>
> [6] Vutukur, S. R., Haugaard, R. L., Huang, J., Busam, B., & Birdal, T. (2024). Alignist: CAD-Informed Orientation Distribution Estimation by Fusing Shape and Correspondences.
>
>
> ----
>
> We thank you for the time invested, insightful questions and provided references, which we believe strengthen the manuscript. We hope our latest revision clarifies your questions and concerns. Please let us know if you have any follow-up / additional questions.
>
> Best regards,
>
> The Authors

---

> ### Author Response · Authors · 2025-11-27
> **Update to reviewer ff67**
>
> We thank you again for your detailed feedback. We have updated our manuscript and added a set of experiments comparing RECON to EquiAdapt [1] as proposed.
>
> ---
>
> Section 5.2 and Figure 6d now include a quantitative comparison with Equivariant Adaptation (EquiAdapt) for test-time canonicalization on the per-class rotated datasets. RECON underperforms EquiAdapt in MNIST by a small margin (90.9% vs EquiAdapt’s 94.5%), but outperforms it in FashionMNIST (82.0% vs 79.4%) and GEOM-QM9 (55.9% vs 52.4%), while keeping the benefits of being a plug-and-play canonicalization layer that is trained without supervision, whereas EquiAdapt’s canonicalization is trained end-to-end using supervision from class labels.
>
> To give further context on where RECON sits relative to symmetry-specialized architectures, we also a comparison against fully equivariant SE(2) ESCNN [2] /SEGNN [3] backbones and equivariance learning methods (Partial G-CNNs [4]) trained from scratch on the same task (Table 3). Notably, RECON stays close to fully E(2) equivariant Steerable CNNs on MNIST, and even surpasses them on FashionMNIST (81.96% vs 78.17%), while consistently outperforming Partial G-CNNs on both datasets. On GEOM-QM9, we underperform fully equivariant networks (SEGNN), but still are able to improve over the pre-trained baseline, IE-AE and EquiAdapt.
>
> ---
>
> We hope this new revision addresses all your concerns and thank you for providing valuable feedback. A brief global summary of all changes across revisions is also provided in our general official comment. Please let us know if you have any follow-up / additional questions.
>
> Best regards,
>
> The Authors
>
> ---
>
> References:
>
> [1] Mondal, A. K., Panigrahi, S. S., Kaba, S.-O., Rajeswar, S., & Ravanbakhsh, S. (2023). Equivariant Adaptation of Large Pretrained Models
>
> [2] Cesa, G., Lang, L., & Weiler, M. (2022). A Program to Build E(N)-Equivariant Steerable CNNs.
>
> [3] Brandstetter, J., Hesselink, R., Pol, E. van der, Bekkers, E. J., & Welling, M. (2022). Geometric and Physical Quantities Improve E(3) Equivariant Message Passing
>
> [4] Learning Partial Equivariances from Data. Romero, David W., and Suhas Lohit.

---

### Official Review · Reviewer_jX6K · 2025-11-02

**Soundness:** 4
**Presentation:** 4
**Contribution:** 2
**Rating:** 8
**Confidence:** 4

**Summary:**

The authors introduce a method for identifying the natural pose of a class in an unsupervised way. In contrast to class-pose methods such as IE-AE, the proposed approach learns a data-aligned canonical. This is achieved using the machinery of IE-AE. Embeddings of IE-AE suggest transformation distributions for dataset instances in connected components. The natural pose is defined as the center of this distribution and obtained using the Frechet mean. The authors argue that a data-aligned canonical is important for various downstream tasks and demonstrate its utility for OOD detection and test time canonicalization.

**Strengths:**

**Originality.** The work appears original.

**Quality.** I rate the quality of this work as good. The work is well motivated, the proposed approach addresses the gap in the literature, and the empirical analysis support the paper claims.

**Clarity.** The organization is good, and the writing is very good.

**Significance.** I rate the significance of this work as good. The work has the potential for broad applicability.

**Weaknesses:**

No notable weaknesses

**Questions:**

At this phase of the review process I do not have clarifying questions.

---

> ### Author Response · Authors · 2025-11-21
> **Response to reviewer jX6K**
>
> We appreciate the positive feedback and thank you for taking the time to carefully read and assess our paper.  We provided a revised manuscript, with all changes highlighted in blue for ease of reference.
>
> In the revision, we mainly focused on incorporating missing relevant literature into the Related work section, and adding additional analyses to address the concerns raised by the other reviewers, e.g., on equivalence class construction, sensitivity to the neighborhood size, and computational overhead. We believe these additions place RECON more clearly within the broader symmetry discovery literature, and provide a more complete picture of its robustness and practical advantages, as well as its limitations and mitigation strategies. Please feel welcome to take a look at the updated manuscript.
>
> We hope that these clarifications and additions reinforce your positive view of the paper and its potential applicability.
>
> Many thanks for your time and effort.
>
> Best regards,
>
> The Authors

---

### Author Response · Authors · 2025-11-27
**Official comment by The Authors**

We would like to thank all reviewers once again for the careful reading, constructive criticism, and the relevant references they provided. We have uploaded a revised version of the manuscript, with changes highlighted in blue for ease of reference. Below we briefly summarize all the changes.

---

**Conceptual clarifications and updated literature**

- We corrected and softened our 3D claim and now position this contribution more precisely with respect to other relevant literature.
- We have expanded the Related work section to incorporate the suggested literature on symmetry discovery via Lie generators and Lie derivatives (LieGAN/LaLiGAN, LieGG and others), as well as related pose-distribution approaches on SO(3), and compared them with our method (Sec. 4).
- We updated the manuscript to use more accurate wording regarding the symmetry discovery task (from discovering “symmetry distributions” to “pose distributions”) and added remarks clarifying the difference with estimating stabilizers (Appx. D.1).

**Robustness, computational analysis, and deep-dive on equivalence class construction**

- We added a more systematic analysis of neighborhood size, prediction error, and hit rates, both in images and molecular conformations. We now report mean and standard deviation over multiple seeds and a range of neighbors, and show that RECON’s symmetry discovery, OOD detection and test-time canonicalization performance is stable across reasonable choices of neighborhood size and runs, among other insights (Appx D.2, D.3).
- We include new quantitative and qualitative analyses on the equivalence class construction, highlighting failure modes and mitigation strategies, making the strengths and limitations of the proposed equivalence class more transparent (Appx. D.3).
- We added a short computational analysis of RECON’s overhead and comments on scaling to larger datasets (Appx D.5).

**New quantitative comparisons with related work in test-time canonicalization**

Several reviewers asked for a clearer picture of how RECON compares to existing methods. In the new revision, we add quantitative comparisons against the main classes of methods that were proposed, namely:
- Equivariant Adaptation (EquiAdapt [1]), as a test-time canonicalization baseline (Table 6d);
- Group-equivariant methods (E(2) Equivariant Steerable CNNs  [2] / SEGNNs [3]); (Table 3)
- Equivariance-learning methods (Partial G-CNNs [4]). (Table 3)

Overall, we find that RECON’s canonicalization consistently improves over the non-canonicalized baseline and provides competitive or better performance than EquiAdapt, while being trained without supervision. In the imaging experiments, our canonicalization consistently outperforms Partial G-CNNs (sometimes by a large gap as in FashionMNIST). Against fully equivariant baselines, RECON comes moderately close in performance (MNIST) and even surpasses them (FashionMNIST), while retaining the benefits of being a plug-and-play layer that can be attached in front of an arbitrary backbone.  On GEOM-QM9, we underperform fully equivariant networks (SEGNN), but still are able to improve over the pre-trained baseline, IE-AE and EquiAdapt. A detailed discussion can be found in Appx F.2.

---

We hope this revision addresses the main concerns raised in the reviews and provides a clearer view of where RECON sits in the growing symmetry discovery literature, as well as its strengths and limitations.

We thank all reviewers and the Area Chair for their time and effort. We greatly appreciate your feedback and believe it strengthens our paper. Please feel welcome to share any additional comments or questions.

Best regards,

The Authors

---
References

[1] Mondal, A. K., Panigrahi, S. S., Kaba, S.-O., Rajeswar, S., & Ravanbakhsh, S. (2023). Equivariant Adaptation of Large Pretrained Models

[2] Cesa, G., Lang, L., & Weiler, M. (2022). A Program to Build E(N)-Equivariant Steerable CNNs.

[3] Brandstetter, J., Hesselink, R., Pol, E. van der, Bekkers, E. J., & Welling, M. (2022). Geometric and Physical Quantities Improve E(3) Equivariant Message Passing

[4] Learning Partial Equivariances from Data. Romero, David W., and Suhas Lohit.

---

### Meta-Review · Area_Chair_bpkG · 2025-12-17

**Summary:**

The submission introduces a method to discover instance-specific pose distributions which enables test-time canonicalization without retraining. Reviewers praised the interesting, effective and theoretically sound ideas, the novelty, potential applicability and writing quality. Reviewers raised concerns about overclaiming contributions due to missing references, the need to know the group a priori, lack of comparisons to some baselines and efficiency.

Initial reviews were mostly positive, except for ff67. The rebuttal was extensive and addressed most concerns and strengthened the submission. Thus, I recommend acceptance.

**Reviewer Concerns:**

The rebuttal included extensive discussion of related work and softening of some claims that ignored prior work. The rebuttal also clarified that while there are methods discovering 3D symmetries, the approach is unique in the sense that it can learn instance-wise symmetry distributions and be used for test-time canonicalization. Extensive analysis of failures modes, efficiency, ablations, and more comparisons against baselines were included. I believe all relevant reviewer concerns were addressed.

**Reviewer Scores:**

ff67 was the only negative review. Their concerns about missing references/overclaims, limitation to known groups, efficiency and variance of results were addressed, so I believe they wold raise their score.

The other reviewers were already positive and their minor concerns were also resolved so they would likely maintain their scores.

---

### Decision · Program_Chairs · 2026-01-26

Accept (Poster)